# Bounds all around: training energy-based models with bidirectional bounds

**Cong Geng, Jia Wang, Zhiyong Gao**
Shanghai Jiao Tong University
{gengcong, jiawang, zhiyong.gao}@sjtu.edu.cn

**Jes Frellsen,**[*] **Søren Hauberg**[*]
Technical University of Denmark
{jefr, sohau}@dtu.dk

## Abstract

Energy-based models (EBMs) provide an elegant framework for density estimation, but they are notoriously difficult to train. Recent work has established links to generative adversarial networks, where the EBM is trained through a minimax game with a variational value function. We propose a bidirectional bound on the EBM log-likelihood, such that we maximize a lower bound and minimize an upper bound when solving the minimax game. We link one bound to a gradient penalty that stabilize training, thereby provide grounding for best engineering practice. To evaluate the bounds we develop a new and efficient estimator of the Jacobi-determinant of the EBM generator. We demonstrate that these developments stabilize training and yield high-quality density estimation and sample generation.

## 1 Energy-based models

Energy-based models (EBMs) are probabilistic models that draw inspiration from physics and have a long history in machine learning (Hopfield, 1982; Hinton and Sejnowski, 1983; Smolensky, 1986). An EBM is specified in terms of an energy function $E_\theta : \mathcal{X} \to \mathbb{R}$ that is parameterized by $\theta$ and defines a probability distribution over $\mathcal{X}$ from the Gibbs density:

$$p_\theta(\mathbf{x}) = \frac{\exp(-E_\theta(\mathbf{x}))}{Z_\theta}, \qquad Z_\theta = \int \exp(-E_\theta(\mathbf{x}))\mathrm{d}\mathbf{x}, \tag{1}$$

where $Z_\theta$ is the normalization constant or partition function. In principle, any density can be described this way for a suitable choice of $E_\theta$. EBMs are typically learned using maximum likelihood estimation (MLE), where we wish to find a value of $\theta$ that minimized the *negative* data log-likelihood:

$$L(\theta) := -\mathbb{E}_{\mathbf{x} \sim p_{\text{data}}(\mathbf{x})}\left[\log p_\theta(\mathbf{x})\right] = \mathbb{E}_{\mathbf{x} \sim p_{\text{data}}}[E_\theta(\mathbf{x})] + \log Z_\theta, \tag{2}$$

where $p_{\text{data}}$ is the data generating distribution.

The fundamental challenge with EBMs for non-standard energy functions is the lack of a closed-form expression for the normalization constant, $Z_\theta$, which hinders exact pointwise density evaluation, sampling and learning. Therefore, these tasks are traditionally performed using approximate methods such as Markov chain Monte Carlo (MCMC, Metropolis et al., 1953; Neal et al., 2011). Similar to the Boltzmann learning rule (Hinton and Sejnowski, 1983; Osogami, 2017), gradient-based learning for MLE involves evaluating the gradient $\nabla_\theta L(\theta) = \mathbb{E}_{\mathbf{x} \sim p_{\text{data}}}[\nabla_\theta E_\theta(\mathbf{x})] - \mathbb{E}_{\mathbf{x} \sim p_\theta}[\nabla_\theta E_\theta(\mathbf{x})]$, where the expectation in the last term necessitates MCMC approximation. Recently, a series of papers (Kim and Bengio, 2016; Kumar et al., 2019; Abbasnejad et al., 2020; Che et al., 2020) have established links between EBMs and *Wasserstein GANs* (WGANs, Arjovsky et al., 2017), such that EBMs can be approximately learned through a minimax game. Xie et al. (2017, 2018c) and Wu et al. (2018) also employed the adversarial game of EBM. This allows for significant improvements in EBM learning over MCMC based methods (Grathwohl et al., 2021).

---

[*]equal contribution

35th Conference on Neural Information Processing Systems (NeurIPS 2021).

**In this paper**, we remark that variational bounds on the value function of the minimax game can be problematic as this is both maximized and minimized. We propose to both upper and lower bound the negative log-likelihood, and alternate between their respective minimization and maximization to alleviate this concern. Evaluation of the bounds requires evaluating the entropy of the generator, and we provide a new efficient estimator. We link the upper bound to the use of gradient penalties which are known to stabilize training. Experimentally, we demonstrate that our approach matches or surpasses state-of-the-art on diverse tasks at negligible performance increase.

### 1.1 Background: Variational bounds and minimax games

Following Grathwohl et al. (2021), we can lower bound the log normalization constant using a proposal (variational) distribution $p_g(\mathbf{x})$ and Jensen's inequality:

$$\log Z_\theta = \log \mathbb{E}_{\mathbf{x} \sim p_g} \left[ \frac{\exp(-E_\theta(\mathbf{x}))}{p_g(\mathbf{x})} \right] \geq \mathbb{E}_{\mathbf{x} \sim p_g} \left[ \log \frac{\exp(-E_\theta(\mathbf{x}))}{p_g(\mathbf{x})} \right] = -\mathbb{E}_{\mathbf{x} \sim p_g}[E_\theta(\mathbf{x})] + H[p_g]$$
(3)

where $H[p_g] = -\mathbb{E}_{p_g}[\log p_g]$ is the (differential) entropy of the proposal distribution. This means that we also obtain a lower bound on the negative log likelihood function given by

$$\mathcal{L}(\theta) := \mathbb{E}_{\mathbf{x} \sim p_{\text{data}}(\mathbf{x})}[E_\theta(\mathbf{x})] - \mathbb{E}_{\mathbf{x} \sim p_g(\mathbf{x})}[E_\theta(\mathbf{x})] + H[p_g] \leq L(\theta).$$
(4)

We note that this bound is tight when $p_g = p_\theta$, since the bound on the log normalization constant (3) can be equivalently expressed as $\log Z_\theta \geq \log Z_\theta - \text{KL}\,(p_g\|p_\theta)$. To tighten the bound, we seek a proposal distribution $p_g$ that maximises $\mathcal{L}(\theta)$, while for MLE we want to find a $\theta$ that minimizes $\mathcal{L}(\theta)$. Accordingly, MLE can be formulated as a minimax game:

$$\min_{E_\theta} \max_{p_g} \{\mathcal{L}(\theta)\} = \min_{E_\theta} \max_{p_g} \left\{ \mathbb{E}_{\mathbf{x} \sim p_{\text{data}}(\mathbf{x})}[E_\theta(\mathbf{x})] - \mathbb{E}_{\mathbf{x} \sim p_g(\mathbf{x})}[E_\theta(\mathbf{x})] + H[p_g] \right\}.$$
(5)

This gives a tractable and MCMC-free approach to EBM learning, assuming we can evaluate the entropy (Grathwohl et al., 2021).

**Our key issue** is that both minimizing and maximizing a lower bound is potentially unstable. By minimizing a lower bound, we run the risk of finding an optimum, where the bound is loosest, rather than where it is informative about the optima of the true objective. In particular, a minima of the lower bound may be $-\infty$, which is rather unhelpful. Good results have been reported from using a lower bound in the minimax game (Dai et al., 2017; Kumar et al., 2019; Grathwohl et al., 2021), but the conceptual issue remains.

**The WGAN loss function** is, as noted by Arjovsky et al. (2017), very similar to the lower bound $\mathcal{L}(\theta)$,

$$L_{\text{WGAN}} = \mathbb{E}_{\mathbf{x} \sim p_{\text{data}}}[E_\theta(\mathbf{x})] - \mathbb{E}_{\mathbf{x} \sim p_g}[E_\theta(\mathbf{x})],$$
(6)

with the missing entropy term being the only difference. Lessons from the WGAN can, thus, be expected to apply to the EBM setting as well. For example, the success of *gradient clipping* (Arjovsky et al., 2017) and *gradient penalties* (Gulrajani et al., 2017) in WGAN training, inspired Grathwohl et al. (2021) to heuristically use gradient penalties in variational inference to stabilize training. Our work will provide justification to such heuristics.

## 2 Approximate minimax games through bidirectional bounds

Instead of solving a minimax game with a lower bounded value function (5), we propose to bound $L(\theta)$ from both above and below:

$$\lfloor L(\theta) \rfloor \leq L(\theta) \leq \lceil L(\theta) \rceil.$$
(7)

With this, we can now follow an optimization strategy of alternating

1. minimize $\lceil L(\theta) \rceil$ with respect to $E_\theta$.
2. maximize $\lfloor L(\theta) \rfloor$ with respect to $p_g$.

This avoids the potential pitfalls of minimizing a lower bound.

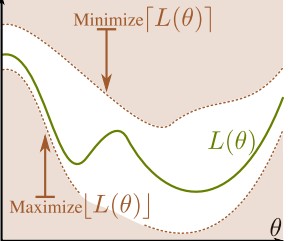

Figure 1: The bidirectional bounds 'sandwich' the negative data log-likelihood.

## 2.1 A lower bound for maximizing $p_g$

We already have a lower bound $\mathcal{L}(\theta)$ on $L(\theta)$ in Equation (4). The first two terms of the bound can readily be evaluated by sampling from $p_{\text{data}}$ and $p_g$, respectively. The entropy of $p_g$ is less straightforward. Following standard practice in GANs, we define $p_g$ through a base variable $\mathbf{z} \sim \mathcal{N}(\mathbf{0}, \mathbf{I})$, which is passed through a 'generator' network $G : \mathbb{R}^d \to \mathbb{R}^D$, i.e. $\mathbf{x} = G(\mathbf{z}) \sim p_g$. This constructs a density over a $d$-dimensional manifold[2] in $\mathbb{R}^D$, which by standard change-of-variables is

$$\log p_g(G(\mathbf{z})) = \log p_0(\mathbf{z}) - \frac{1}{2} \log \det(\mathbf{J}_{\mathbf{z}}^{\mathsf{T}}\mathbf{J}_{\mathbf{z}}), \tag{8}$$

where $\mathbf{J}_{\mathbf{z}} \in \mathbb{R}^{D \times d}$ is the Jacobian of $G$ at $\mathbf{z}$. This assume that the Jacobian exist, i.e. that the generator $G$ has at least one derivative. The entropy of $p_g$ is then

$$H[p_g] = -\mathbb{E}_{\mathbf{x} \sim p_g} [\log p_g(\mathbf{x})] = H[p_0] + \mathbb{E}_{\mathbf{z} \sim p_0} \left[ \frac{1}{2} \log \det(\mathbf{J}_{\mathbf{z}}^{\mathsf{T}}\mathbf{J}_{\mathbf{z}}) \right]. \tag{9}$$

The entropy of $p_0$ trivially evaluates to $H[p_0] = {}^{d}\!/_{2}(1 + \log(2\pi))$. To avoid an expensive evaluation of the log-determinant-term, we note that it is easily bounded from below:

$$\frac{1}{2} \log \det(\mathbf{J}_{\mathbf{z}}^{\mathsf{T}}\mathbf{J}_{\mathbf{z}}) = \frac{1}{2} \sum_{i=1}^{d} \log s_i^2 \geq d \log s_1, \tag{10}$$

where $s_d \geq \ldots \geq s_1$ are the singular values of the Jacobian $\mathbf{J}_{\mathbf{z}}$. Note that in general, this bound is not tight. With this, we have our final lower bound:

$$L(\theta) \geq \lfloor L(\theta) \rfloor = \mathbb{E}_{\mathbf{x} \sim p_{\text{data}}(\mathbf{x})}[E_\theta(\mathbf{x})] - \mathbb{E}_{\mathbf{x} \sim p_g(\mathbf{x})}[E_\theta(\mathbf{x})] + H[p_0] + \mathbb{E}_{\mathbf{z} \sim p_0}[d \log s_1(\mathbf{z})]. \tag{11}$$

## 2.2 An upper bound for minimizing $E_\theta$

There are many ways to arrive at an upper bound for the log-likelihood. Due to the strong ties between the lower bound (4) and the WGAN objective, we take inspiration from how WGANs are optimized. The original WGAN paper (Arjovsky et al., 2017) noted optimization instabilities unless *gradient clipping* was applied. Gulrajani et al. (2017) noted that this clipping could be avoided by adding a *gradient penalty*, such that the loss become

$$L_{\text{WGAN-GP}} = L_{\text{WGAN}} + \lambda \mathbb{E}_{\hat{\mathbf{x}} \sim g(\mathbf{x})} \left[ (\|\nabla_{\hat{\mathbf{x}}} E_\theta(\hat{\mathbf{x}})\|_2 - 1)^2 \right], \tag{12}$$

where $\hat{x}$ is a uniform interpolation between real data and generated samples. This is generally acknowledged as a good way to train a WGAN (Gulrajani et al., 2017). We observe that this loss is similar to our lower bound, but the gradient penalty adds a positive term, such that one could speculate that we might have an upper bound. With this intuition in hand, we prove the following statement in the supplementary material.

**Theorem 1** *Suppose $f : \mathcal{X} \to \mathbb{R}$ is L-Lipschitz continuous, $g(x)$ is a probability density function with finite support, then there exists constants $M, m \geq 0$ and $p \geq 1$ such that:*

$$\log \mathbb{E}_{\mathbf{x} \sim g(\mathbf{x})} [f(\mathbf{x})] - \mathbb{E}_{\mathbf{x} \sim g(\mathbf{x})} [\log f(\mathbf{x})] \leq M (\mathbb{E}_{\mathbf{x} \sim g(\mathbf{x})} [|\nabla_{\mathbf{x}} \log f(\mathbf{x})|^p] + m)^{1/p}. \tag{13}$$

Note that if $\mathbb{E}_{\mathbf{x} \sim g(\mathbf{x})} [|\nabla_{\mathbf{x}} \log f(\mathbf{x})|^p] + m \geq 1$, the bound in Equation (13) can be simplified by dropping the power $^1\!/_p$,

$$\log \left[ \mathbb{E}_{\mathbf{x} \sim g(\mathbf{x})} [f(\mathbf{x})] \right] - \mathbb{E}_{\mathbf{x} \sim g(\mathbf{x})} [\log f(\mathbf{x})] \leq M \mathbb{E}_{\mathbf{x} \sim g(\mathbf{x})} [|\nabla_{\mathbf{x}} \log f(\mathbf{x})|^p] + Mm. \tag{14}$$

For our model, we set $f(\mathbf{x}) = \frac{\exp(-E_\theta(\mathbf{x}))}{p_g(\mathbf{x})}$ and $g(\mathbf{x}) = p_g(\mathbf{x})$. Empirically, we observe the bound simplification in Equation (14) holds through-out most of the training, and use the following upper bound:

$$\lceil L(\theta) \rceil = \mathcal{L}(\theta) + M \mathbb{E}_{\mathbf{x} \sim p_g(\mathbf{x})} [|\nabla_{\mathbf{x}} E_\theta(\mathbf{x}) + \nabla_{\mathbf{x}} \log p_g(\mathbf{x})|^p] + Mm. \tag{15}$$

For a given choice of $f(\mathbf{x})$, we can view $m$ as a constant and the term $Mm$ can be ignored during optimization. This upper bound can directly be interpreted as the lower bound (4) plus a gradient penalty, albeit one of a different form than the traditional WGAN penalty, which is derived purely from a regularization perspective. Our upper bound can, thus, be seen as a justification of the regularization from a maximum likelihood perspective.

---

[2]Note that Eq. 8 is only the density *on* the spanned manifold, and that the density is zero *off* the manifold.

**Bound tightness** When $p_\theta = p_g$, we have that $|\nabla_\mathbf{x} E_\theta(\mathbf{x}) + \nabla_\mathbf{x} \log p_g(\mathbf{x})|^p = 0$. In Theorem 1, $m$ is a constant related to the Lipschitz constant of $\log f(\mathbf{x})$ satisfying $|\nabla_\mathbf{x} \log f(\mathbf{x})|^p \leq |\nabla_\mathbf{y} \log f(\mathbf{y})|^p + m$ for all $\mathbf{x}, \mathbf{y}$ (see proof for details). When $p_\theta = p_g$ we also have $|\nabla_\mathbf{x} \log f(\mathbf{x})|^p = 0$, such that $m = 0$. Our upper bound is then $\lceil L(\theta) \rceil = \mathcal{L}(\theta) = L(\theta)$, and hence it is tight.

## 3 Numerical evaluation of the bounds

**Evaluating the lower bound** To evaluate the lower bound in Equation (11), we need the smallest singular value of the Jacobian $\mathbf{J} = \partial_\mathbf{z} G(\mathbf{z})$. Recall that this singular value satisfy $s_1 = \|\mathbf{J}\mathbf{v}_{\min}\|_2 = \min_{\mathbf{v} \neq \mathbf{0}} \frac{\|\mathbf{J}\mathbf{v}\|_2}{\|\mathbf{v}\|_2}$. We can then evaluate the singular value by finding $\mathbf{v}_{\min}$ with an iterative optimization algorithm, where we opt to use the celebrated single-vector *LOBPCG algorithm* (Knyazev, 1998). This method performs an iterative minimization of the generalized Rayleigh quotient,

$$\rho(\mathbf{v}) := \frac{\mathbf{v}^\intercal \mathbf{J}^\intercal \mathbf{J} \mathbf{v}}{\mathbf{v}^\intercal \mathbf{v}}, \tag{16}$$

which converges to $\mathbf{v}_{\min}$. The gradient of $\rho(\mathbf{v})$ is proportional to $r = \mathbf{J}^\intercal \mathbf{J} \mathbf{v} - \rho(\mathbf{v})\mathbf{v}$. To avoid computing the Jacobian $\mathbf{J}$, we use Jacobian-vector products, which can be efficiently evaluated using automatic differentiation. To compute $\mathbf{J}^\intercal \mathbf{J} \mathbf{v}$, we use the following trick (in `pytorch`-notation):

$$\mathbf{J}^\intercal \mathbf{J} \mathbf{v} = ((\mathbf{J}\mathbf{v})^\intercal \mathbf{J})^\intercal = \nabla_\mathbf{z} \left( (\mathbf{J}\mathbf{v})^\intercal \texttt{.detach()} \cdot G(\mathbf{z}) \right)^\intercal. \tag{17}$$

The optimal learning rate for this iterative scheme can be found by maximizing the Rayleigh quotient (16). Finally, we follow the suggestions of Knyazev (2001) to improve numerical stability and accelerate convergence, which we omit here for brevity.

**Evaluating the upper bound** There are two challenges when evaluating Equation (15). The first is to compute $\nabla_\mathbf{x} \log p_g(\mathbf{x})$, where we empirically found that existing methods (Shi et al., 2018; Li and Turner, 2018) were too inefficient for our needs. To evaluate the term $\mathbb{E}_{\mathbf{x} \sim p_g(\mathbf{x})} [|\nabla_\mathbf{x} E_\theta(\mathbf{x}) + \nabla_\mathbf{x} \log p_g(\mathbf{x})|^p]$, we further loosen the bound

$$
\begin{aligned}
|\nabla_{G(\mathbf{z})} E_\theta(G(\mathbf{z})) + \nabla_{G(\mathbf{z})} \log p_g(G(\mathbf{z}))|^p &\leq \frac{|\nabla_{G(\mathbf{z})} E_\theta(G(\mathbf{z}))\mathbf{J_z} + \nabla_{G(\mathbf{z})} \log p_g(G(\mathbf{z}))\mathbf{J_z}|^p}{s_1^p} \\
&\leq \frac{|\nabla_{G(\mathbf{z})} E_\theta(G(\mathbf{z}))\mathbf{J_z} + \nabla_\mathbf{z} \log p_g(G(\mathbf{z}))|^p}{s_1^p}.
\end{aligned}
\tag{18}
$$

where $s_1$ is the smallest singular value of $\mathbf{J_z}$. Detailed derivations are in the supplementary material. If we choose $p = 2$, then we can use Hutchinson's estimator (1989):

$$|\nabla_\mathbf{x} E_\theta(\mathbf{x})\mathbf{J_z} + \nabla_\mathbf{z} \log p_g(G(\mathbf{z}))|^2 = \mathbb{E}_\mathbf{v} \left[ (\nabla_\mathbf{x} E_\theta(\mathbf{x})\mathbf{J_z}\mathbf{v} + \nabla_\mathbf{z} \log p_g(G(\mathbf{z}))\mathbf{v})^2 \right], \tag{19}$$

where $\mathbf{v} \sim \mathcal{N}(\mathbf{0}, \mathbf{I}_d)$. This is easily evaluated using automatic differentiation.

The second challenge is to evaluate $\log p_g(\mathbf{x})$ which needs the Jacobian of the generator $G(\mathbf{z})$ as dictated by Equation (8). Here, we opt to use our entropy estimator as described above. We could alternatively use Hutchinson's estimator as proposed by Kumar et al. (2020). Experimentally we do not observe much difference between these two estimators.

## 4 Related work

In machine learning, there has been a long-standing interest in EBMs dating back to Hopfield networks (Hopfield, 1982), Boltzmann machines (Hinton and Sejnowski, 1983; Ackley et al., 1985) and restricted Boltzmann machines (Smolensky, 1986; Hinton, 2002), see e.g. reviews in the works by LeCun et al. (2006) and Scellier (2020). Learning and evaluation of these models are difficult since the normalization constant cannot be efficiently evaluated. MLE-based learning, such as the Boltzmann learning rule, relies on expensive MCMC sampling to estimate the gradient, and more advanced MCMC methods are used to reliably estimate the normalization constant (see e.g. Salakhutdinov and Murray, 2008; Grosse et al., 2013; Burda et al., 2015; Frellsen et al., 2016). For images, MCMC-based learning has been used to learn non-deep EBMs of both textures (Zhu et al., 1998; Zhu and

Mumford, 1998) and natural images (Xie et al., 2015, 2016). Learning algorithms that avoid the costly MCMC approximation have been heavily investigated. For instance, Hinton (2002) proposed k-step Contrastive Divergence (CD-k) to approximate the negative phase log-likelihood gradient, and Hyvärinen (2005) proposed an alternative method to train non-normalized graphical models using score matching. Deep versions of EBMs have subsequently been proposed, such as deep belief networks (Hinton et al., 2006) and deep Boltzmann machines (Salakhutdinov and Hinton, 2009).

In recent years, there have been renewed interest in deep generative EBMs, particularly for image generation. Grathwohl et al. (2021) gave an excellent overview of the current developments and distinct drawbacks of three classes of learning methods, which we summarize here: (1) MLE methods with MCMC sampling are slow and unstable, and while (2) score matching based methods are comparable faster, they are also unstable and do not work with discontinuous nonlinearities. (3) Noise-contrastive estimation (Gutmann and Hyvärinen, 2010) do not have these drawback, but it does not scale well with the data dimensionality.

Our proposed method belongs to a fourth class of algorithms that sidestep the costly MCMC sampling by using a simultaneously learned generator or variational distribution. Kim and Bengio (2016) proposed using a generator function and an adversarial-like training strategy similar to ours. They update the generator using the same lower bound as us, but their entropy approximation is quite different. Furthermore, their method does not have a gradient penalty like our upper bound when optimizing the energy function. Consequently, the energy function needs to be explicitly designed to prevent it from growing to infinity, limiting its potential.

The method proposed by Zhai et al. (2016) plays a min-max game to jointly optimize the energy function and generator using the same lower bound as us. However, their method relies on a specific designed bounded multi-modal energy function, which limits its potential. Furthermore, their approximation of the generator entropy comes with no theoretical guarantees, and their regularisation of the energy function does not constitute an upper bound.

Dai et al. (2017) proposed an adversarial learning framework for jointly learning the energy function and a generator. They considered two approaches to maximising the generator entropy: One, which maximizes the entropy by minimizing the conditional entropy using a variational upper bound, and another, which makes isotropic Gaussian assumptions for the data, which is not suitable for high-dimensional data. Kumar et al. (2019) and Abbasnejad et al. (2019) also consider adversarial learning but different approaches to estimating the entropy of the generator. Kumar et al. (2019) estimated the entropy through its connection to mutual information, but they need an additional network to measure the entropy term. Abbasnejad et al. (2019) maximized the entropy by approximating the generator function's Jacobian log-determinant. However, their method is impractical in high dimensions as the Jacobian is computationally expensive. Han et al. (2019) use an adversarial learning strategy in their divergence triangle loss, but their training mechanism is radically different from ours, and they rely on an extra encoder for learning the generator.

Xie et al. (2018a,b, 2021a,b) proposed *cooperative learning* of the energy function and a generator. However, the cooperative learning approach relies on MCMC or Langevin dynamics to draw samples from the EBM, which is expensive and difficult to tune. Like our work, VERA by Grathwohl et al. (2021) avoids the use of MCMC. VERA plays a min-max game and uses variational inference to approximate the gradient of the entropy term, which is different from ours. Furthermore, VERA uses a gradient penalty as a regularizer for the energy function, which is a heuristic unlike our upper bound, and their method is memory-consuming, and the hyperparameters are difficult to adjust.

## 5   Experiments

To demonstrate the efficiency of our energy-based model with bidirectional bounds (EBM-BB) we compare against a range of methods that represent state-of-the-art. As representatives of the GAN literature, we consider *deep convolutional GANs (DCGANs)* by Radford et al. (2016), *spectrally normalized GANS (SNGANs)* by Miyato et al. (2018) and the *WGAN with zero-centered gradient penalty (WGAN-0GP)* by Thanh-Tung et al. (2019). As representatives of EBMs, we consider the *maximum entropy generators (MEG)* by Kumar et al. (2019) and the *variational entropy regularized approximate maximum likelihood (VERA)* estimator (Grathwohl et al., 2021). As representatives of CoopNets, we consider two similar methods (Xie et al., 2018a, 2021a) and *EBM-VAE* (Xie et al., 2021b). We also consider *NCSN* by Song and Ermon (2019) and *DDPM* by Ho et al. (2020)

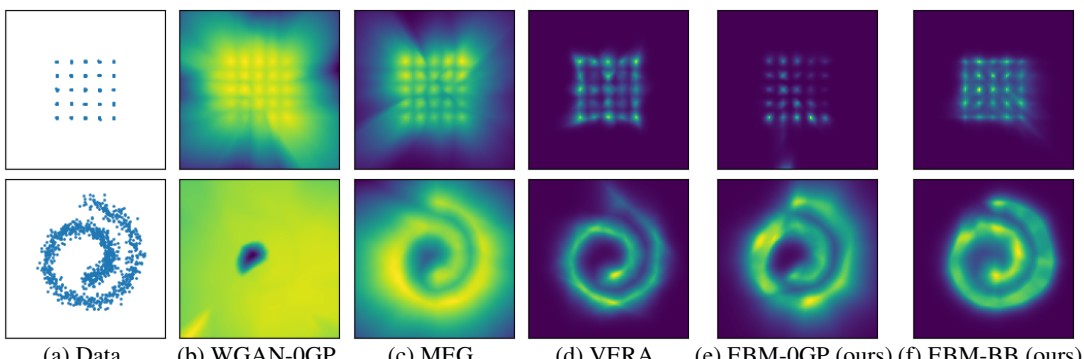

| (a) Data | (b) WGAN-0GP | (c) MEG | (d) VERA | (e) EBM-0GP (ours) | (f) EBM-BB (ours) |

Figure 2: Density estimation on the 25-Gaussians and swiss-roll datasets.

as representatives of a score matching model and a diffusion model respectively. To investigate the influence of our derived upper bound, we introduce another baseline in which we replace our upper bound with a zero-centered gradient penalty. We denote the resulting model *EBM-0GP*. Our implementation is available at https://github.com/gengcong940126/EBM-BB.

## 5.1 Training details

On toy data and MNIST, all models are realized with multi-layer perceptrons (MLPs), while for natural images, we use the convolutional architecture from `StudioGAN` (Kang and Park, 2020). Details of the network architecture are given in the supplementary material. All experiments are conducted on a single 12GB NVIDIA Titan GPU using a `pytorch` (Paszke et al., 2017) implementation. To improve training, we use a positive margin for our energy function to balance the bounds. Specifically, we let

$$\lceil L(\theta)\rceil = \lfloor L(\theta)\rfloor + \left[M\mathbb{E}_{\mathbf{x}\sim p_g(\mathbf{x})}\left[|\nabla_{\mathbf{x}}E_\theta(\mathbf{x}) + \nabla_{\mathbf{x}}\log p_g(\mathbf{x})|^p\right] - \zeta\right]_+, \tag{20}$$

where $[\cdot]_+ = \max(0, \cdot)$ is the usual hinge. In most experiments, $\zeta = 1$ as this allows us to apply the simplified bound in Equation (15). But in general we recommend starting with $\zeta = 0$ and trying successively larger values to stabilize training.

## 5.2 Toy data

Figure 2 shows estimated densities on the 25-Gaussians and swiss-roll datasets using both our methods, WGAN-0GP and two baselines. We observe that WGAN's discriminator is not suitable as a density estimator. This is unsurprising as WGAN is not supposed to provide a density estimate. MEG and EBM-0GP can have inaccurate density information in some edge and peripheral areas. This may be due to insufficient or excessive maximization of entropy since the zero-centered gradient penalty is not a principled objective for maximum

Table 1: Number of captured modes and KL divergence between the real and generated distributions.

| Model | Modes↑ | KL↓ |
|---|---|---|
| DCGAN | $392 \pm 7.4$ | $8.012 \pm 0.056$ |
| SNGAN | $441 \pm 39.0$ | $2.755 \pm 0.033$ |
| WGAN-0GP | $\mathbf{1000 \pm 0.0}$ | $0.048 \pm 0.003$ |
| MEG | $\mathbf{1000 \pm 0.0}$ | $0.042 \pm 0.004$ |
| VERA | $989 \pm 9.0$ | $0.152 \pm 0.037$ |
| EBM-0GP(ours) | $\mathbf{1000 \pm 0.0}$ | $\mathbf{0.039 \pm 0.003}$ |
| EBM-BB (ours) | $\mathbf{1000 \pm 0.0}$ | $0.045 \pm 0.003$ |

likelihood. VERA and our EBM-BB can learn a sharp distribution, but our method is more stable in some inner regions of the 25-Gaussians and edge regions of the swiss-roll.

## 5.3 MNIST

**Mode counting** Mode collapse is a frequently occurring phenomenon in which the generator function maps all latent input to a small number of points in the observation space. In particular, GANs are plagued by this problem. Since the generator is trained to maximize the entropy of the generated distribution, several EBM-based models have been shown to capture all the modes of the data distribution. To empirically verify that our model also captures a variety of modes in the data distribution, we follow the test procedure of Kumar et al. (2019). We train our generative

model on the StackedMNIST dataset, a synthetic dataset created by stacking MNIST on different channels. The true total number of modes is $1,000$, and they are counted using a pretrained MNIST classifier. The KL divergence is calculated empirically between the generated mode distribution and the data distribution. The results appear in Table 1. As expected, GAN-based methods suffer from mode collapse except for WGAN-0GP. All the EBM-based methods capture all or nearly all the modes of the data distribution. Our model EBM-0GP captures all modes and report the smallest KL measures. It worth mentioned that WGAN-0GP obtains comparable results, which was also observed by Grathwohl et al. (2021).

**Entropy Estimation**  In Figure 3, we explore the quality of our entropy estimator measured on MNIST (LeCun, 1998). We use networks with fully connected layers as the Jacobian is then easily derived in closed-form, giving us a ground truth for the entropy. We compare our estimator (red) with the ground truth (black) and Hutchinson's estimator (blue), as was proposed by Kumar et al. (2020). Finally, as we only run our iterative estimator for a few steps, we also compare with an estimator that uses the smallest singular value computed with high precision (green). We observe that Hutchinson's estimator is reasonable close to the ground truth but provides an upper bound, making it inapplicable for our lower

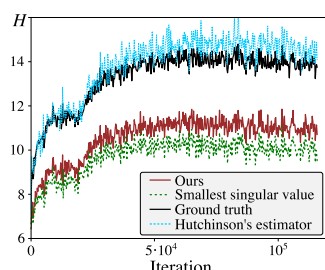

Figure 3: Entropy estimators.

bound. We further observe a noticeable gap between our estimator and the high precision singular value estimator. Finally, we see that all estimators follow roughly the same trend, which suggests that our estimator provides a suitable target for optimization.

## 5.4  Natural Images

**Image generation**  The data studied thus far are simple and perhaps do not challenge our upper bound and entropy estimator. We, therefore, train our model on the standard benchmark $32 \times 32$ CIFAR-10 (Krizhevsky et al., 2009) dataset and the $64 \times 64$ cropped ANIMEFACE[3] dataset, which both represent a significant increase in complexity. Similar to recent work on GANs (Miyato et al., 2018), we report Inception Score (IS), Fréchet Inception Distance (FID) scores and two $F_\beta$ scores (Sajjadi et al., 2018). We compare with competitive GANs, EBM baselines, CoopNets, NCSN and DDPM (Table 3). All GANs and EBMs are reproduced using the same network architecture (see supplements for details) except DCGAN and IGEBM, where we respectively borrow the results from StudioGAN (Kang and Park, 2020) and the original implementation (Du and Mordatch, 2019). We use the DCGAN (Radford et al., 2016) network architecture for CIFAR-10 and a Resnet architecture (Kang and Park, 2020) for ANEMIFACE. Parameters are chosen as in the original papers. For VERA (Grathwohl et al., 2021), we were unable to reproduce the reported performance, so we choose hyper-parameters according to an extensive grid search. Like the original paper, we choose the entropy weight to be $0.0001$. For CoopNets and NCSN, we report the results from the original papers. For DDPM, we used a public available implementation.[4]

From Table 3, we see that our EBM-BB model is the best performing EBM on CIFAR-10, though it is surpassed by NCSN and DDPM. This is not surprising as NCSN and DDPM focus on sample quality rather than optimising a data likelihood. Further note that sampling NCSN and DDPM is significantly more expensive than our method. On ANIMEFACE our method is highly competitive. Figures 4 and 5 show samples from different models. For ANIMEFACE, our method generates more diverse samples in terms of face parts than the baselines. All models predominantly generate female faces. This suggests that while our approach captures more density modes than the baselines, all approaches still misses several modes. We further draw attention to several corrupt samples generated by VERA, despite extensive parameter search. For CIFAR-10, we observe no immediate differences in the generative capabilities between models.

**Capacity usage**  For the proposal distribution $p_g$ to be adaptive, the associated generator network $G(\mathbf{z})$ should be able to use as much of its available capacity as possible. To compare different methods, we consider an implicit measure of capacity usage $\mathcal{C}_\mathbf{z}$ that locally measures the intrinsic dimensionality of $G(\mathbf{z})$. In particular, we use the anisotropy index (Wang and Ponce, 2021) that, for

---

[3]https://www.kaggle.com/splcher/animefacedataset

[4]https://github.com/rosinality/denoising-diffusion-pytorch

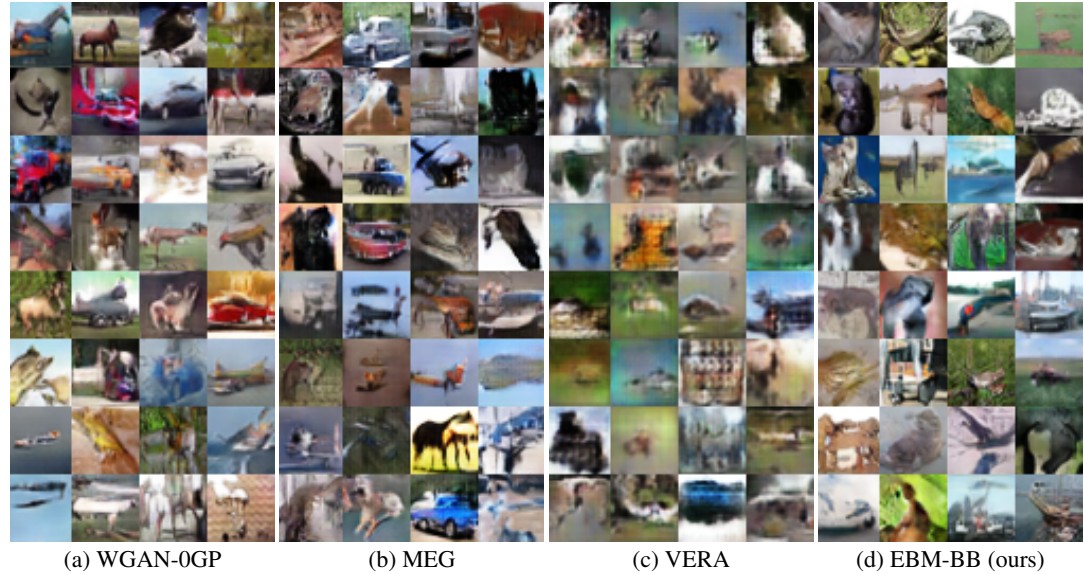

| (a) WGAN-0GP | (b) MEG | (c) VERA | (d) EBM-BB (ours) |

Figure 4: Generated samples on CIFAR-10 with our method and various methods.

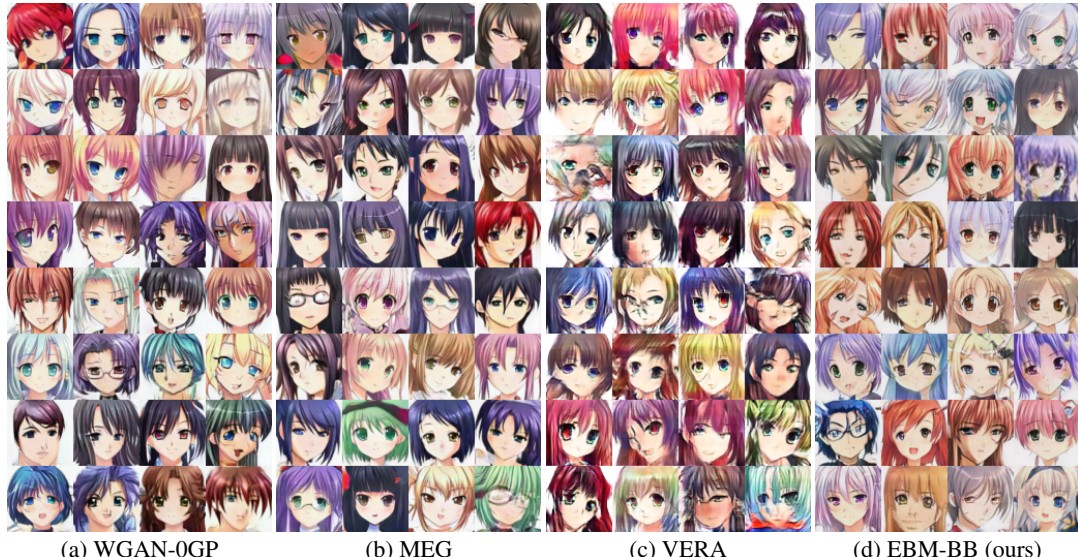

| (a) WGAN-0GP | (b) MEG | (c) VERA | (d) EBM-BB (ours) |

Figure 5: Generated samples on ANIMEFACE with our method and other generative models

a given $\mathbf{z}$, measure the standard deviation of the norm of the directional derivative of $G(\mathbf{z})$ along the input dimensions, i.e.

$$\mathcal{C}_{\mathbf{z}} = \mathrm{std}\left(\{\|\mathbf{J}_{\mathbf{z}}\mathbf{e}_i\|\}_{i=1\ldots d}\right), \tag{21}$$

where $\mathrm{std}(\cdot)$ computes the standard deviation of the input, and $\mathbf{e}_i$ denotes the $i^{\mathrm{th}}$ standard basis vector. A small $\mathcal{C}_{\mathbf{z}}$ indicates that the different input dimensions contribute equally to the output of $G$, which imply good capacity usage. We measure the mean value of $\mathcal{C}_{\mathbf{z}}$ for $\mathbf{z} \sim \mathcal{N}(\mathbf{0}, \mathbf{I}_d)$ and report mean and standard deviation of the result running for several times in Table 2.

Table 2: Anisotropy indices ($\downarrow$).

| Model | CIFAR-10 | ANIMEFACE |
|---|---|---|
| WGAN-0GP | $1.0823 \pm 0.005$ | $2.4366 \pm 0.129$ |
| MEG | $0.9467 \pm 0.009$ | $2.3600 \pm 0.142$ |
| VERA | $3.9694 \pm 0.055$ | $2.2929 \pm 0.2074$ |
| EBM-0GP (ours) | $1.0688 \pm 0.0226$ | $3.4560 \pm 0.0521$ |
| EBM-BB (ours) | $\mathbf{0.9431 \pm 0.0191}$ | $\mathbf{1.8016 \pm 0.1345}$ |

We observe that most models, with the exception of VERA, perform well on CIFAR-10, while on ANIMEFACE there is more diversity. In both cases, EBM-BB has the best capacity usage. The large difference between EBM-0GP and EBM-BB on

Table 3: Comparison in terms of FID, Inception Score, $F_8$ score (weights recall higher than precision) and $F_{1/8}$ score (weights precision higher than recall).

| Model | Inception↑ | FID↓ | $F_8$↑ | $F_{1/8}$↑ |
|---|---|---|---|---|
| | | CIFAR-10 | | |
| DCGAN | 6.64 | 49.03 | 0.795 | 0.83 |
| WGAN-0GP | $7.24 \pm 0.035$ | $29.31 \pm 0.185$ | $0.92 \pm 0.010$ | $0.95 \pm 0.010$ |
| CoopNets (2018a) | 6.55 | 36.4 | - | - |
| CoopNets (2021a) | - | 33.61 | - | - |
| EBM-VAE | 6.65 | 36.2 | - | - |
| IGEBM | 6.78 | 38.2 | - | - |
| MEG | $6.62 \pm 0.243$ | $34.55 \pm 1.145$ | $0.88 \pm 0.001$ | $0.92 \pm 0.010$ |
| VERA | $5.06 \pm 0.555$ | $66.38 \pm 6.635$ | $0.58 \pm 0.080$ | $0.79 \pm 0.005$ |
| NCSN | $8.87 \pm 0.12$ | 25.32 | - | - |
| DDPM | **9.03** | **7.76** | **0.98** | **0.99** |
| EBM-0GP (ours) | $6.90 \pm 0.032$ | $35.42 \pm 0.582$ | $0.90 \pm 0.004$ | $0.93 \pm 0.002$ |
| EBM-BB (ours) | $7.45 \pm 0.014$ | $28.63 \pm 0.290$ | $0.93 \pm 0.001$ | $0.95 \pm 0.008$ |
| | | ANIMEFACE | | |
| WGAN-0GP | $2.22 \pm 0.030$ | $9.76 \pm 0.674$ | $\mathbf{0.95 \pm 0.005}$ | $\mathbf{0.98 \pm 0.005}$ |
| MEG | $2.20 \pm 0.020$ | $9.31 \pm 0.007$ | $\mathbf{0.95 \pm 0.005}$ | $\mathbf{0.98 \pm 0.001}$ |
| VERA | $2.15 \pm 0.001$ | $41.00 \pm 1.072$ | $0.515 \pm 0.078$ | $0.78 \pm 0.013$ |
| DDPM | 2.18 | **8.81** | 0.94 | **0.98** |
| EBM-0GP (ours) | $\mathbf{2.26 \pm 0.017}$ | $20.53 \pm 0.524$ | $0.889 \pm 0.008$ | $0.909 \pm 0.019$ |
| EBM-BB (ours) | $\mathbf{2.26 \pm 0.005}$ | $12.75 \pm 0.045$ | $0.94 \pm 0.001$ | $0.96 \pm 0.005$ |

Table 4: AUROC↑, AUPRC↑ and FPR80↓ for OOD detection for 'train / test' datasets.

| Model | CIFAR-10 / SVHN | | | CIFAR-10 / CIFAR-100 | | | ANIMEFACE / Bedroom | | |
|---|---|---|---|---|---|---|---|---|---|
| | AUROC↑ | AUPRC↑ | FPR80↓ | AUROC↑ | AUPRC↑ | FPR80↓ | AUROC↑ | AUPRC↑ | FPR80↓ |
| WGAN-0GP | 0.8 | 0.83 | 0.39 | 0.54 | 0.55 | 0.77 | 0.51 | 0.48 | 0.72 |
| MEG | 0.79 | 0.81 | 0.42 | 0.52 | 0.53 | 0.8 | 0.56 | 0.53 | 0.77 |
| VERA | 0.62 | 0.64 | 0.64 | 0.51 | 0.51 | 0.79 | 0.60 | 0.525 | 0.6 |
| EBM-0GP (ours) | 0.66 | 0.69 | 0.67 | **0.64** | **0.63** | **0.64** | **0.67** | **0.63** | **0.53** |
| EBM-BB (ours) | **0.88** | **0.86** | **0.1997** | 0.53 | 0.52 | 0.7765 | 0.53 | 0.505 | 0.83 |

ANIMEFACE suggest that our proposed upper bound helps increase the entropy of the proposal distribution $p_g$.

**Out-of-distribution detection** As EBMs are density estimators, they should, in principle, assign low likelihood to out-of-distribution (OOD) observations. OOD detection performance can then be seen as a proxy for the quality of the estimated density. To test how well our model fares in this regard, we follow previous work (Hendrycks and Gimpel, 2017; Hendrycks et al., 2019; Alemi et al., 2018; Choi et al., 2018; Ren et al., 2019; Havtorn et al., 2021) and report the threshold independent evaluation metrics of Area Under the Receiver Operator Characteristic (AUROC↑), Area Under the Precision Recall Curve (AUPRC↑) and False Positive Rate at 80% (FPR80↓), where the arrow indicates the direction of improvement of the metrics. The results are reported in Table 4. Each column of the table takes the form 'In-distribution / Out-of-distribution' in reference to the training and test set, respectively. We observe that on CIFAR-10 / SVHN, EBM-BB is the top performer. On CIFAR-10 / CIFAR-100 the overall performance degraded significantly, EBM-0GP fared noticeable better than other models. The overall degradation may be caused by the strong similarity between CIFAR-10 and CIFAR-100. For ANIMEFACE / Bedroom a similar situation occurs, even if these datasets are highly dissimilar, but we observe EBM-0GP is much better than other models.

No clear winner can be found from this study. We note that our method consistently performs well, but surprisingly, so does WGAN-0GP even if it is not a density estimator. As OOD detection is a task that comes with many subtle pitfalls (Havtorn et al., 2021), we suggest that these results should be taken with a grain of salt even if our model performs well.

**Running time** Table 5 gives an overview of the time required to train the proposed models and the baselines. All models were trained on a single 12GB Titan GPU. We observe that our model should be trained with a lower learning rate, and therefore may increase the total number of epochs.

## 6 Discussion

The first observation in our work is that current methods for training energy-based models (EBMs) interchangeably minimize and maximize a lower bound. As this may be a potential source of training instability, we propose to bound the negative log-likelihood from above and below and switch between bounds when minimizing and

Table 5: Total training time.

| Model | Iterations | Runtime |
|-------|-----------|---------|
| CIFAR-10 | | |
| WGAN-0GP | 100000 | 4h |
| MEG | 100000 | 4.5h |
| VERA | 200000 | 10h |
| EBM-0GP | 200000 | 20h |
| EBM-BB | 200000 | 21h |
| ANIMEFACE | | |
| WGAN-0GP | 100000 | 13h |
| MEG | 100000 | 20h |
| VERA | 200000 | 44h |
| EBM-0GP | 100000 | 38h |
| EBM-BB | 100000 | 40h |

maximizing. The lower bound, $\lfloor L(\theta) \rfloor$, is similar to existing ones, but we provide a new algorithm for its realization. Unlike past work, this algorithm does not need additional networks and does not rely on hard-to-tune parameters; our only parameter controls the number of iterations for the Jacobian approximation and represents a trade-off between the accuracy of the bound and computational budget. The upper bound, $\lceil L(\theta) \rceil$, is new to the literature but similar to the common regularization practice of introducing gradient penalties. To the best of our knowledge, this is the first time that gradient penalties have been derived from the perspective of bounding the log-likelihood. It is rewarding that current better engineering practice can be justified from a probabilistic perspective. Empirically, we find our proposed model generally performs as well or better than some of the current state-of-the-art on a variety of tasks. The evidence suggests that our bidirectional bounds allow the generator to increase its entropy and capacity usage. We see this both directly and through improved sample quality on diverse datasets.

**Limitations** The current drawbacks of the method are mainly three concerns. First, we find that a smaller-than-usual learning rate helps our model. While we expect this to be a matter of implementation rather than a more profound concern, the current implication is that training is approximately twice that of a EBM baseline. Second, our bounds rely on computing the smallest singular value of the Jacobian of the generator. We provide an efficient implementation of a method for this task, but it needs to run for several iterations to guarantee convergence. In practice, we stop after a fixed, low number of iterations, which we found to work well, but technically this violates the bound. This seems to be an unavoidable aspect of our approach, but we find that the benefits significantly compensate for this issue. The availability of a fast (approximate) entropy estimator that does not require additional networks and parameters is highly valuable when training EBMs. Finally, for the upper bound, we need an estimate of the volume of support $M$ of the proposal distribution $p_g$. We do not have a viable method for this, and, in practice, we treat $M$ as a hyper-parameter. We have not found it difficult to tune this parameter, but it nonetheless constitutes a limitation.

**Negative societal impact** All high-capacity generative models carry the risk of being used for misinformation, and our model is no exception. The value of EBMs over GANs is that they come with a likelihood function, which is more valuable in data analysis, than e.g. for creating deepfakes.

## Acknowledgments and Disclosure of Funding

This work was funded in part by the Novo Nordisk Foundation through the Center for Basic Machine Learning Research in Life Science (NNF20OC0062606). It also received funding from the European Research Council (ERC) under the European Union's Horizon 2020 research, innovation programme (757360), National science foundation of China under grant 61771305 and Shanghai Municipal Science and Technology Major Project (2021SHZDZX0102). JF was supported in part by the Novo Nordisk Foundation (NNF20OC0065611) and the Independent Research Fund Denmark (9131-00082B). SH was supported in part by a research grant (15334) from VILLUM FONDEN. The authors are grateful to anonymous reviewers and the handling area chair for valuable discussions and feedback on an early version of this manuscript.

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
