# Bounds all around: training energy-based models with bidirectional bounds — Supplementary Material —

**Cong Geng, Jia Wang, Zhiyong Gao**
Shanghai Jiao Tong University
{gengcong, jiawang, zhiyong.gao}@sjtu.edu.cn

**Jes Frellsen,** * **Søren Hauberg** *
Technical University of Denmark
{jefr, sohau}@dtu.dk

## A Theoretical part

### A.1 Proof of Theorem 1

**Proof**

$$
\log \left[ \mathbb{E}_{\mathbf{x} \sim g(\mathbf{x})} \left[ f(x) \right] \right] - \mathbb{E}_{\mathbf{x} \sim g(\mathbf{x})} \left[ \log f(x) \right] = \log \int f(x) g(x) dx - \int (\log f(x)) g(x) dx
$$

$$
= \log \left[ f(\eta) \int g(x) dx \right] - \int (\log f(x)) g(x) dx = \log f(\eta) \int g(x) dx - \int (\log f(x)) g(x) dx
$$

$$
= \int \left[ \log f(\eta) - \log f(x) \right] g(x) dx = \int g(x) \int_0^1 \frac{d \left[ \log f(t\eta + (1-t)x) \right]}{dt} dt dx
$$

$$
= \int g(x) \int_0^1 \nabla_{\tilde{x}} \log f(\tilde{x})(\eta - x) dt dx
$$

$$
\leq \int g(x) \left( \int_0^1 |\nabla_{\tilde{x}} \log f(\tilde{x})|^p dt \right)^{\frac{1}{p}} \left( \int_0^1 |\eta - x|^q dt \right)^{\frac{1}{q}} dx
$$

$$
\leq \int g(x) |\eta - x| \left( \int_0^1 |\nabla_{\tilde{x}} \log f(\tilde{x})|^p dt \right)^{\frac{1}{p}} dx
$$

$$
\leq \left( \int g(x) \int_0^1 |\nabla_{\tilde{x}} \log f(\tilde{x})|^p dt dx \right)^{\frac{1}{p}} \left( \int g(x) |\eta - x|^q dx \right)^{\frac{1}{q}}, \tag{1}
$$

*where $\tilde{x} = t\eta + (1-t)x$. The second equation is derived by mean value theorem for definite integrals. The first inequality is derived by Holder's inequality, so $p, q \geq 1$ and $\frac{1}{p} + \frac{1}{q} = 1$. Because $g(x)$ has finite support, there exists an $M \geq 0$ that satisfying: $|\eta - x| \leq M$, then we can get:*

$$
\log \left[ \mathbb{E}_{\mathbf{x} \sim g(\mathbf{x})} \left[ f(x) \right] \right] - \mathbb{E}_{\mathbf{x} \sim g(\mathbf{x})} \left[ \log f(x) \right] \leq M \left( \int g(x) \int_0^1 |\nabla_{\tilde{x}} \log f(\tilde{x})|^p dt dx \right)^{\frac{1}{p}}
$$

$$
\leq M \left( \int g(x) |\nabla_{\hat{x}} \log f(\hat{x})|^p dx \right)^{\frac{1}{p}}, \tag{2}
$$

*where $\hat{x} = t_0 \eta + (1 - t_0)x$ for a $t_0$ ($0 \leq t_0 \leq 1$) using the mean value theorem. Because $f(x)$ is $L$-Lipschitz continuous, then $\log f(x)$ is also Lipschitz continuous, so there exists an $m \geq 0$ satisfying*

$$
|\nabla_{\hat{x}} \log f(\hat{x})|^p \leq |\nabla_x \log f(x)|^p + m, \text{ } for \forall x, \tag{3}
$$

35th Conference on Neural Information Processing Systems (NeurIPS 2021).

*so we can get*

$$\log\left[\mathbb{E}_{\mathbf{x}\sim g(\mathbf{x})}\left[f(x)\right]\right] - \mathbb{E}_{\mathbf{x}\sim g(\mathbf{x})}\left[\log f(x)\right] \leq M\left(\int g(x)|\nabla_x \log f(x)|^p dx + m\right)^{\frac{1}{p}} \tag{4}$$

$$\leq M(\mathbb{E}_{\mathbf{x}\sim g(\mathbf{x})}\left[|\nabla_x \log f(x)|^p\right] + m)^{\frac{1}{p}}.$$

### A.2  Proof of Eq. (18)

**Proof** *For a vector $u \in R^{1\times n}$ and a matrix $J \in R^{n\times m}$, we have*

$$|uJ|_2^2 = uJJ^T u^T = \text{tr}(JJ^T u^T u) \tag{5}$$

$$\geq \lambda(JJ^T)\,\text{tr}(u^T u),$$

*where $\lambda$ is the smallest eigenvalue of $JJ^T$. The inequality holds because $JJ^T$ is a real symmetric matrix and $u^T u$ is positive semidefinite (Neudecker, 1992). Then we set $J = \mathbf{J_z}, u = \nabla_{\mathbf{x}}E_\theta(G(\mathbf{z})) + \nabla_{G(\mathbf{z})}\log p_g(G(\mathbf{z}))$, we can obtain:*

$$|\nabla_{\mathbf{x}}E_\theta(G(\mathbf{z}))\mathbf{J_z} + \nabla_{G(\mathbf{z})}\log p_g(G(\mathbf{z}))\mathbf{J_z}|_2^2 \geq \lambda(\mathbf{J_z J_z^T})\,\text{tr}(u^T u) \tag{6}$$

$$= \lambda(\mathbf{J_z J_z^T})|u|_2^2 = \lambda(\mathbf{J_z J_z^T})|\nabla_{\mathbf{x}}E_\theta(G(\mathbf{z})) + \nabla_{G(\mathbf{z})}\log p_g(G(\mathbf{z}))|_2^2$$

*Because $\lambda(\mathbf{J_z J_z^T})$ is the square of the smallest singular value of $\mathbf{J_z}$, which we represent by $s_1$. So we can obtain:*

$$|\nabla_{\mathbf{x}}E_\theta(G(\mathbf{z})) + \nabla_{G(\mathbf{z})}\log p_g(G(\mathbf{z}))|_2 \leq \frac{|\nabla_{\mathbf{x}}E_\theta(G(\mathbf{z}))\mathbf{J_z} + \nabla_{G(\mathbf{z})}\log p_g(G(\mathbf{z}))\mathbf{J_z}|_2}{s_1}. \tag{7}$$

*This proves Eq. 18.*

## B  Model Architecture

In order apply the change-of-variables formula to get a density for the generator, we assume that $G : \mathbb{R}^d \to \mathbb{R}^D$ spans an immersed $d$-dimensional manifold in $\mathbb{R}^D$. This assumption place some restrictions on the architecture of the generator neural network.

The governing assumption is that the Jacobian of $G$ exist and has full rank. Existence is ensured as long as the chosen activation functions have at least one derivative almost everywhere. Smooth activations naturally satisfy this assumption, but it is worth noting that e.g. the ReLU activation function has a single point where the derivative is not defined. As long as the linear map preceding the activation is not degenerate, then the non-smooth region has measure zero, and the change-of-variables technique still applies.

We cannot guarantee that the Jacobian has full rank through clever choices of neural architectures. However, we note that one requirement is that no hidden layer may have dimensionality below the $d$ dimensions of the latent space. This is a natural requirement for the generator anyway. In our model, we aim to maximize the entropy of the generator, which encourages the generator to create as diverse samples as possible. In practice this ensures that the Jacobian has full rank as a degenerate Jacobian implies a reduction of entropy. Note that this is not a theoretical guarantee against degenerate Jacobians during optimization, but in practice we have at no point experienced problems in this regard.

### B.1  Practical experimental settings

For the toy and MNIST datasets, we use multi-layer perceptrons (MLPs) networks, while for CIFAR-10 and ANIMEFACE datasets, we use a DCGAN network and Resnet architecture, respectively.

## C  Training Details

In Table 6, we specify the hyperparameters used when training our models for each dataset. We choose $p = 2$ in Eq (20) for our implementation. We normalize the data to be in [-1, 1] and do not

Table 1: The network architecture trained for toy datasets

| Operation | Input | Output |
|---|---|---|
| Energy | | |
| Linear, PReLU | 2 | 100 |
| Linear, PReLU | 100 | 100 |
| Linear | 100 | 1 |
| Generator | | |
| Linear, PReLU, BN | 2 | 100 |
| Linear, PReLU, BN | 100 | 100 |
| Linear | 100 | 2 |

Table 2: The network architecture trained for MNIST dataset

| Operation | Input | Output |
|---|---|---|
| Energy | | |
| Linear, PReLU | 2352 | 2000 |
| Linear, PReLU | 2000 | 1000 |
| Linear, PReLU | 1000 | 500 |
| Linear, PReLU | 500 | 250 |
| Linear, PReLU | 250 | 250 |
| Linear | 250 | 1 |
| Generator | | |
| Linear, BN, PReLU | 128 | 500 |
| Linear, BN, PReLU | 500 | 1000 |
| Linear, BN, PReLU | 1000 | 2000 |
| Linear, Tanh | 2000 | 2352 |

Table 3: The network architecture for CIFAR-10 dataset

| Operation | Kernel | Strides | Channels | Output size |
|---|---|---|---|---|
| Energy | | | | |
| Conv2D, LReLU | 3 | 1 | 64 | 32 |
| Conv2D, LReLU | 4 | 2 | 64 | 16 |
| Conv2D, LReLU | 3 | 1 | 128 | 16 |
| Conv2D, LReLU | 4 | 2 | 128 | 8 |
| Conv2D, LReLU | 3 | 1 | 256 | 8 |
| Conv2D, LReLU | 4 | 2 | 256 | 4 |
| Conv2D, LReLU | 3 | 1 | 512 | 4 |
| Flatten | | | | 8192 |
| Linear | | | | 1 |
| Generator | | | | |
| Linear | | | | 8192 |
| Reshape | | | 512 | 4 |
| ConvTranspose2D, BN, ReLU | 4 | 2 | 256 | 8 |
| ConvTranspose2D, BN, ReLU | 4 | 2 | 128 | 16 |
| ConvTranspose2D, BN, ReLU | 4 | 2 | 64 | 32 |
| Conv2D, Tanh | 3 | 1 | 3 | 32 |

Table 4: The energy network architecture for ANIMEFACE dataset

| Operation | Kernel | Strides | Channels | Output size |
|---|---|---|---|---|
| (ResBlock0) | | | | |
| Left: Conv2D, BN, LReLU | 3 | 1 | 64 | 64 |
| Conv2D | 3 | 1 | 64 | 64 |
| AvgPool2D | 2 | 2 | 64 | 32 |
| Right: AvgPool2D, BN | 2 | 2 | 3 | 32 |
| Conv2D | 1 | 1 | 64 | 32 |
| Overall: Add | | | 64 | 32 |
| (ResBlock1) | | | | |
| Left: BN, LReLU,Conv2D | 3 | 1 | 128 | 32 |
| BN, LReLU,Conv2D | 3 | 1 | 128 | 32 |
| AvgPool2D | 2 | 2 | 128 | 16 |
| Right: BN, Con2D | 1 | 1 | 128 | 32 |
| AvgPool2D | 2 | 2 | 128 | 16 |
| Overall: Add | | | 128 | 16 |
| (ResBlock2) | | | | |
| Left: BN, LReLU,Conv2D | 3 | 1 | 256 | 16 |
| BN, LReLU,Conv2D | 3 | 1 | 256 | 16 |
| AvgPool2D | 2 | 2 | 256 | 8 |
| Right: BN, Con2D | 1 | 1 | 256 | 16 |
| AvgPool2D | 2 | 2 | 256 | 8 |
| Overall: Add | | | 256 | 8 |
| (ResBlock3) | | | | |
| Left: BN, LReLU,Conv2D | 3 | 1 | 512 | 8 |
| BN, LReLU,Conv2D | 3 | 1 | 512 | 8 |
| AvgPool2D | 2 | 2 | 512 | 4 |
| Right: BN, Con2D | 1 | 1 | 512 | 8 |
| AvgPool2D | 2 | 2 | 512 | 4 |
| Overall: Add | | | 512 | 4 |
| (ResBlock4) | | | | |
| Left: BN, LReLU,Conv2D | 3 | 1 | 1024 | 4 |
| BN, LReLU,Conv2D | 3 | 1 | 1024 | 4 |
| Right: BN, Con2D | 1 | 1 | 1024 | 4 |
| Overall: Add | | | 1024 | 4 |
| LReLU | | | 1024 | 4 |
| Sum | | | | 1024 |
| Linear | | | | 1 |

use dequantization. During training we augment only using random horizontal flips. For cifar10 we used the official train-test split from PyTorch, and for animeface we used a 85:15 train-test split. For our upper bound, we set $\frac{M}{s_1^2} = \frac{0.001}{z_{dim}}$, where $s_1$ is the smallest singular value of Jacobian and $z_{dim}$ is the latent dimension. We observe this setting can get a satisfying generation for all datasets. If we replace the $z_{dim}$ with $\|v\|_2^2$ in high-dimensional data, where $v$ is a random vector sampled in Eq 19, it will further improve the generation. For out-of-distribution detection and capacity usage, we set $\frac{M}{s_1^2} = \frac{0.1}{z_{dim}}$, because if we increase this value, it will help the density estimation of the energy function. We also observe that the network's design affect performance. For example, removing batch normalization in the energy function can stabilize training on the Animeface dataset. The relationship between such design decisions in the context of EBMs should be explored in future work.

## References

H. Neudecker. A matrix trace inequality. *Journal of mathematical analysis and applications*, 166(1): 302–303, 1992.

Table 5: The generator network architecture for ANIMEFACE dataset

| Operation | Kernel | Strides | Channels | Output size |
|---|---|---|---|---|
| Linear | | | | 16384 |
| Reshape | | | 1024 | 4 |
| (ResBlock0) | | | | |
| Left: BN, ReLU, NN-Upsampling, Conv2D | 3 | 1 | 512 | 8 |
| BN, ReLU, Conv2D | 3 | 1 | 512 | 8 |
| Right:  NN-Upsampling, Conv2D | 1 | 1 | 512 | 8 |
| Overall:  Add | | | 512 | 8 |
| (ResBlock1) | | | | |
| Left: BN, ReLU, NN-Upsampling, Conv2D | 3 | 1 | 256 | 16 |
| BN, ReLU, Conv2D | 3 | 1 | 256 | 16 |
| Right:  NN-Upsampling, Conv2D | 1 | 1 | 256 | 16 |
| Overall:  Add | | | 256 | 16 |
| (ResBlock2) | | | | |
| Left: BN, ReLU, NN-Upsampling, Conv2D | 3 | 1 | 128 | 32 |
| BN, ReLU, Conv2D | 3 | 1 | 128 | 32 |
| Right:  NN-Upsampling, Conv2D | 1 | 1 | 128 | 32 |
| Overall:  Add | | | 128 | 32 |
| (ResBlock3) | | | | |
| Left: BN, ReLU, NN-Upsampling, Conv2D | 3 | 1 | 64 | 64 |
| BN, ReLU, Conv2D | 3 | 1 | 64 | 64 |
| Right:  NN-Upsampling, Conv2D | 1 | 1 | 64 | 64 |
| Overall:  Add | | | 64 | 64 |
| BN, ReLU, Conv2D, Tanh | 3 | 1 | 3 | 64 |

Table 6: Selection of most important hyper-parameters and their setting.

| Datasets | Optimization | Learning rate | Batch size | Iterations/Epochs | latent dim |
|---|---|---|---|---|---|
| Toy | Adam(0.0,0.9) | 2e-4 | 200 | 150000 | 2 |
| MNIST | Adam(0.0,0.9) | 2e-4 | 64 | 60(epochs) | 128 |
| CIFAR-10 | Adam(0.0,0.999) | 5e-5 | 64 | 200000 | 128 |
| ANIMEFACE | Adam(0.0,0.999) | 5e-5 | 64 | 100000 | 128 |