# OpenReview forum: "Bounds all around: training energy-based models with bidirectional bounds"
_NeurIPS.cc/2021/Conference — NeurIPS 2021 Poster_

### Official Review · Reviewer_rhEU · 2021-07-07

**Rating:** 6
**Confidence:** 3

**Summary:**

This work proposes a new method for density estimation using Energy-based model. Classical variational bounds replace the negative log-likelihood by a computable lower bound, and phrase the maximum likelihood problem as a minimax game on this lower bound. A possible algorithm to solve the problem is then to alternate between minimization and maximization of this lower bound (with respect to different variables). Although this approach has shown good performance, the authors claim that the concept of minimizing such a lower bound can lead to instabilities. In particular, they claim that the WGAN objective tends to be unstable to optimize, and that while clipping the gradient in this case solves the instability issue, the origin of the instability lies in minimizing an upper bound. To circumvent this issue, the current work constructs both an upper and a lower bound to the negative log-likelihood, and alternate between minimizing the upper bound and maximizing the lower bound. The upper bound is inspired by what is done in WGAN, i.e., by adding a gradient penalty. The algorithm is tested on several benchmarks, and is shown to be rather competitive compared to existing algorithms.

**Limitations And Societal Impact:**

The authors adequately addressed the limitations and potential negative societal impact

**Main Review:**

The authors claim that the reason for adding gradient penalty in WGAN is to stabilize the training. However, the reason for this penalty comes from the problem formulation of learning a distribution by minimizing the Wasserstein distance. By Kantorovich duality, this leads to a minimax game with Lipschitz constraint, which is enforced by using either gradient clipping or gradient penalty. The use of gradient penalty in WGAN is hence very well motivated. In contrast, the use of gradient penalty in the current framework seems much more abstract than in WGAN.

The proposed algorithm bears strong similarities with WGAN with gradient penalty. The empirical evaluation actually shows quite similar performance, although the proposed method takes about twice more time to train (Table 5).
Hence, I don’t see the benefit of this approach compared to WGAN, both in theory and in practice.


Typos:

l.54 tends

Thm 1 there exist


**Time Spent Reviewing:**

3h

---

> ### Author Response · Authors · 2021-08-10
> **Reply**
>
> Thanks for the feedback. We fully agree that gradient penalty in WGAN is principled and not a 'work around' as implied by line 55. Our poor wording was due to other EBMs use gradient penalties heuristically. For example, Grathwohl et al. (2021) use a gradient penalty without justification, while Kumar et al. (2019) provide a lengthy discussion of the empirical importance of gradient penalties (again, no formal justification is provided).
> Our work provides the missing justification. Our objective was, thus, not to criticize WGANs in this regard, but rather to point to heuristic uses in the EBM literature.
>
> Similarly, we did not aim to say that WGANs are unstable to train, but rather that EBM training can be unstable and that gradient penalties/clipping are one of the reasons WGAN are stable.
>
> In both these cases we admit that our communication was less than stellar and we will rephrase accordingly.
>
> We, however, disagree that our approach lacks benefit compared to WGANs. Loosely speaking, our approach has the benefits of the WGAN alongside the benefits of being an EBM. In terms of theory, our approach (unlike the WGAN) has a well-defined likelihood function. This is paramount for data analysis tasks, where the WGAN is often difficult to apply. We show how some of the strong sides of the WGAN translates to EBM, which is nontrivial. Here we take inspiration from the gradient penalization from the WGAN, but note that we end up with a different term in our bound.
> In terms of practical benefits, this is also evident if one try to use the WGAN discriminator as an energy function (and hence density estimator). The example below on the toy data of Fig. 2 quite clearly shows that the WGAN discriminator is not suitable as a density estimator. This is unsurprising as WGAN is not supposed to provide a density estimate, and we only show this to demonstrate a practical benefit of our approach over the WGAN.
>
> [Link: density estimation results](https://ibb.co/X5DfZxZ)

---

### Official Review · Reviewer_xLGS · 2021-07-12

**Rating:** 5
**Confidence:** 4

**Summary:**

The authors propose to sandwich the EBM log-likelihood with newly introduced upper and lower bounds, with which a minimax objective is formed to train EBMs.
Specifically, based on the variational lower bound from Grathwohl et al. (2021), the authors approximate the entropy term therein to form the presented lower bound, and then adding a gradient-penalty term mimicking WGAN to form the new upper bound.
The lower and upper bounds are optimized wrt different groups of parameters, as, intuitively, both minimizing and maximizing a lower bound wrt the same group of parameters is potentially unstable (this is the main motivation for this submission).

**Limitations And Societal Impact:**

Yes.

**Main Review:**

Originality. Several techniques used to develop the upper/lower bounds are believed sound and novel; they might be useful to audiences from different domains. As for related works, important discussions associated with popular methods for training EBMs (e.g., [1,2]) are believed missing, as well as the empirical comparisons.

Quality. Minor flaws may exist in some presented techniques: (1) Eq. (8) may not hold for general GAN generators; a GAN generator is not an injective function in general, which means the matrix J_z^T J_z may not be invertible. Please elaborate on that. (2) How to guarantee that the condition in Line 86 is satisfied in practice? Please add the related analysis. (3) Is Eq (15) guaranteed to be an upper bound of the log-likelihood?

Clarity. The submission is well-written and easy to follow.

Significance. The training of EBMs is a popular research direction that is of great interest to the ML community. However, I am not fully convinced by the presented method, because of the missing comparisons with [1,2] and the minor flaws mentioned above.

Other comments:

In Line 49-50, why "both minimizing and maximizing a lower bound is potentially unstable"? Also, what's the ''conceptual issue''? These are not clear in the current submission.

Is Eq. (10) "the new and efﬁcient estimator of the Jacobi-determinant of the EBM generator," mentioned in the Abstract? That contribution is not clearly stated.

Since the lower bound in Eq. (11) is not tight, discussions on the gap are necessary. Also, how expensive is the evaluation of Eq. (11)?

I am concerned about the training stability. How stable is the training empirically?

It seems Eqs. (20) and (15) show different upper bounds respectively. Which one is actually used in the experiment? Why?

In the experiments, does the proposed method perform better than the popular methods for training EBMs [1,2]?

Minors:
1. In Line 12, the abbreviation ''EMBs" is a typo.
2. In Line 53, the statement "the missing entropy term being the only difference" may not be right, since the function E_{\theta}(x) in Eq. (6) must be a 1-Lipschitz function.

[1] Yang Song and Stefano Ermon. Generative modeling by estimating gradients of the data distribution. In Advances in Neural Information Processing Systems, pp. 11895–11907, 2019.
[2] Jonathan Ho, Ajay Jain, and Pieter Abbeel. Denoising diffusion probabilistic models. Advances in Neural Information Processing Systems, 33, 2020.

**Time Spent Reviewing:**

3

---

> ### Author Response · Authors · 2021-08-10
> **Reply**
>
> Thank you for the careful review. We section our replies in order of perceived importance.
>
> ## Missing baselines
>
> We are happy to add the requested baselines, NCSN [1] and DDPM [2]. For the rebuttal, we used public code ([NSCN](https://github.com/ermongroup/ncsn), [DDPM](https://github.com/lucidrains/denoising-diffusion-pytorch)) for these papers on *cifar10* and *animeface*. The code for NCSN was too memory-consuming to run on the *animeface* using 2 GPUs in $64 \times 64$ resolution, so only *cifar10* results are reported below. Here we used the default parameters except for *batch size* and *number of iterations*, which we matched with the existing methods of the paper for a fair comparison. Note that the numbers reported below are *not* on par with those reported in the corresponding papers (reverting all parameters to their defaults did not change this observation).
>
> |   Model   |              IS             |              FID             |            $F_8$            |      $F_{\frac{1}{8}}$      |
> |-----------|-----------------------------|------------------------------|-----------------------------|-----------------------------|
> | Cifar10   |                             |                              |                             |                             |
> |    NCSN   |       7.20 $\pm$ 0.183      |      62.43 $\pm$ 10.020      |       0.86 $\pm$ 0.015      |       0.81 $\pm$ 0.020      |
> |    DDPM   |       7.32 $\pm$ 0.105      |       35.70 $\pm$ 0.470      |       0.89 $\pm$ 0.005      |       0.93 $\pm$ 0.009      |
> |    Ours   | $\textbf{7.45 $\pm$ 0.014}$ | $\textbf{28.63 $\pm$ 0.290}$ | $\textbf{0.93 $\pm$ 0.001}$ | $\textbf{0.95 $\pm$ 0.008}$ |
> | Animeface |                             |                              |                             |                             |
> |    DDPM   | $\textbf{2.41 $\pm$ 0.005}$ |      27.34 $\pm$ 0.065     |      0.77 $\pm$ 0.003     |      0.92 $\pm$ 0.003     |
> |    Ours   |      2.26 $\pm$ 0.005     | $\textbf{12.75 $\pm$ 0.045}$ | $\textbf{0.94 $\pm$ 0.001}$ | $\textbf{0.96 $\pm$ 0.005}$ | |
>
> We note that our approach is better in terms of FID and $F_{\beta}$ scores, while DDPM do better in terms of the IS score on animeface.
>
> We also provide qualitative results:
>
> [Link: cifar10 results](https://ibb.co/yWfq8sW)
>
> [Link: animeface results](https://ibb.co/j50CDnj)
>
> Note how DDPM [2] produce samples from the *animeface* dataset where elementary face attributes such as mouth and nose are mostly missing.
>
> Furthermore, we observe that the generating time is significantly larger for these papers than for the existing baselines: on *cifar10*, NCSN [1] is about 3000 times slower than our method, while DDPM [2] is around 700 times as slow.
>
> Finally, we note that these methods follow a very different philosophy from ours. NCSN models $\nabla_{x}\log p(x)$ rather than $\log p(x)$, while DDPM is a diffusion model that is properly normalised (i.e. the normalisation constant $Z$ is known) and it can be learned using variational inference.
>
> ## Analysis
>
> - Regarding Eq. 8 then we cannot guarantee that any neural network $G: \mathbb{R}^d \rightarrow \mathbb{R}^D$ is injective, but note that $G$ only needs to be injective locally for Eq. 8 to hold. That is, Eq. 8 holds if the Jacobian of $G$ has rank $d$ everywhere, which is true as long as the weight matrices associated with $G$ are full rank. This question can also be studied probabilistically, where it is e.g. well-known that if the Jacobian is modeled with independent Gaussian entries, then $J^T J$ is invertible with probability one if $d < D$. In practice, we have yet to observe a singular Jacobian (keep in mind that we repeatedly compute the smallest singular value of $J$, so we would have observed the problem if it were to appear). That being said, we acknowledge that these consideration should be discussed in the paper.
> - Regarding the condition in Line 86, we apply a `hinge trick' described in section 5.1. This ensures that the condition $\mathbb{E}_{x \sim g(x)} \left[\vert\nabla_x \log f(x)\vert^p\right]+m \geq 1$. In practice, we observe that this is almost always satisfied without the hinge trick. Note that by using the hinge trick, we also guarantee that Eq. 15 is an upper bound of the negative log-likelihood.
>
> ## Conceptual issue
>
> In the paper, we tried to argue that minimizing a lower bound is conceptually problematic, but it seems we were not fully convincing (we admit that this argument is not fully fleshed out in the paper). By minimizing a lower bound, we run the risk of finding an optimum, where the bound is loosest, rather than where it is informative about the optima of the true objective. In particular, a minima of the lower bound may be $-\infty$, which is rather unhelpful. In our setting, Eq. 6 can become infinite if the energy distribution collapse to a delta-function. This is a direct consequence of minimizing a lower bound. The WGAN literature often makes a note of this problem. Our solution is to instead work with an upper bound, which mimic gradient penalty regularization.
>
>
> ## Details of the bound and training
>
> - Equation 10 is our new and efﬁcient estimator of the Jacobi-determinant of the EBM generator. This use a single-vector LOBPCG algorithm to find the smallest singular value of the Jacobian (Sec. 3).
> - We find training to be quite stable with a behavior that is similar to the training of WGANs.
> - We cannot provide a complete answer to your question about the gap induced by our singular value-based lower bound. We find that the answer is data-dependent. If the LOBPCG is run to convergence we are guaranteed to have a lower bound, so the question concerns the gap introduced by using finite steps (two, in practice). Figure 3 studies the question to some degree on MNIST, where we find that it is always a lower bound. Note that the algorithm only needs to perform Jacobian-vector products which is inexpensive.

---

> > ### Comment · Reviewer_xLGS · 2021-08-25
> > **Re: Reply**
> >
> > Thanks for the detailed response.
> >
> > Regarding the concern associated with Eq (8), I don't think I am persuaded. In most neural networks, a well-trained weight matrix is definitely not Gaussian distributed, and it often becomes low-rank quickly during training. Once a near-zero s1 is reached, the optimization would become unstable; perhaps the last term in Eq (11) would alleviate this issue.
> >
> > How to prove that Eq (15) is an upper bound of the negative log-likelihood in Eq (2)?
> >
> > The performance of both baseline methods NCSN/DDPM is worse by a large margin than that reported in the original papers. Are these baselines trained to converge?

---

> > > ### Author Response · Authors · 2021-08-27
> > > **Follow-up**
> > >
> > > Thanks for engaging in the discussion; this gives valuable feedback on the paper.
> > >
> > > ## Rank of the Jacobian matrix
> > >
> > > We did not mean to say that the weights become Gaussian, but rather point to this being one instance where one can make a mathematical statement about the rank of the weight matrix. In Eq. 5 we see that the overall objective include a wish to *maximize* the entropy of $p_g$. If a weight matrix become degenerate, then by Eq. 9, the entropy become minus infinity. In practice, the entropy is approximated by the last term of Eq. 11 (that you point to) which indeed encourage the the weight matrices to have full rank. In the case of a degenerate weight matrix, the approximate entropy also become minus infinity. Does this alleviate your concern?
> > >
> > > ## Proof that Eq. 15 is an upper bound of the negative log-likelihood in Eq. 2
> > >
> > > If $\mathbb{E}_{\mathbf{x} \sim g(\mathbf{x})}\left[\left|\nabla_\mathrm{x} \log f(\mathrm{x})\right|^{p}\right]+m \geq 1$ and $p\geq 1$, then we can drop the power $\frac{1}{p}$ in Eq. 13 to get Eq. 14:
> > >
> > > $$\log E_{x \sim g(x)}[f(x)]-E_{x \sim g(x)}[\log f(x)]  \leq M\left(E_{x \sim g(x)}[|\nabla_x \log f(x)|^p]+m \right)^{1/p} \leq M\left(E_{x \sim g(x)}\left[|\nabla_x \log f(x)|^{p}\right]+m\right)= M E_{x \sim g(x)}\left[\left|\nabla_{x} \log f(x)\right|^p\right]+Mm$$
> > >
> > > From Eqs. 2-4 we have:
> > >
> > > $$L(\theta) - \mathcal{L}(\theta)=\left(E_{x \sim p_{data}}\left[E_{\theta}(x)\right]+\log Z_{\theta}\right)-\left(E_{x \sim p_{data}(x)}\left[E_{\theta}(x)\right]-E_{x \sim p_{g}(x)}\left[E_{\theta}(x)\right]+H\left[p_{g}\right]\right)=\log Z_{\theta}-\left(-E_{x \sim p_{g}}\left[E_{\theta}(x)\right]+H\left[p_{g}\right]\right)=\log E_{x \sim p_{g}}\left[\frac{\exp \left(-E_{\theta}(x)\right)}{p_{g}(x)}\right] - E_{x \sim p_{g}}\left[\log \frac{\exp \left(-E_{\theta}(x)\right)}{p_{g}(x)}\right]$$
> > >
> > > Thus if we set $f(\mathbf{x})=\frac{\exp \left(-E_{\theta}(\mathrm{x})\right)}{p_{g}(\mathrm{x})}$ and $g(\mathrm{x})=p_{g}(\mathrm{x})$ in Eq. 14, we obtain $$L(\theta) - \mathcal{L}(\theta)=\log E_{x \sim p_g}\left[\frac{\exp \left(-E_{\theta}(x)\right)}{p_g(x)}\right] - E_{x \sim p_g}\left[\log \frac{\exp \left(-E_{\theta}(x)\right)}{p_{g}(x)}\right] \leq  M E_{x \sim p_g(x)}[|\nabla_x E_{\theta}(x)+\nabla_x \log p_g(x)|^p]+M m$$. This proves that Eq. 15  is an upper bound of Eq. 2 under the stated assumptions. Did this clarify the derivation?
> > >
> > > ## Baseline results
> > >
> > > It is correctly observed that the performance of both baseline methods (NCSN and DDPM) is worse than reported in the original papers. We also noted this in our previous response, but we apologize for not providing further details. When we run the two methods, we observe that they are very sensitive to the networks' architectures and parameter settings. The results we reported for DDPM in the rebuttal, we obtained using the default parameters of a [reference implementation of DDPM by Phil Wang](https://github.com/lucidrains/denoising-diffusion-pytorch). Afterwards, we tried the exact parameters of the original paper, then we obtained the published results on Cifar-10. We find that DDPM can outperform our approach on Cifar10, but that the algorithm is highly sensitive to hyperparameters (see details below). This is not surprising since these models focus on sample quality rather than optimising data likelihood.
> > >
> > > For NCSN, we use their official parameters, except that we set the batch size to 64 as their method consume more memory than ours. Like ref. [1], we found the method to be quite unstable. Because generation is so slow, we chose the best model in 100,000, 150,000 and 200,000 iterations and obtained the results reported in our previous answer.
> > >
> > > For DDPM, we considered two public available PyTorch implementations. We used the default parameters' settings in the [implementation by Phil Wang](https://github.com/lucidrains/denoising-diffusion-pytorch) for 500,000 iterations (Config1). For this implementation, we found that if we change the U-Net structure and parameter settings, the results change a lot (Config2). We have also experimented with the [implementation by Kim Seonghyeon](https://github.com/rosinality/denoising-diffusion-pytorch). With this implementation, we used the model parameters as specified in the paper and obtained comparable results (Config3). For Animeface, there are no default settings in the paper, so for the Phil Wang implementation, we use default settings (Config4), and for the Kim Seonghyeon implementation, we use the default setting for the CelebA dataset (Config5). The results for the different configurations are reported in the table below.
> > >
> > > For our method, we use a very common DCGAN network for Cifar10 and a ResNet architecture for the Animeface dataset that is less memory consuming (we have about ten times fewer parameters than DDPM). Finally, we want to note that in architecture experimentation, we have found that removing the batch normalization in our energy function can further improve our FID score on the Animeface dataset as shown in the table below.
> > >
> > >
> > > |      Config     |   IS  |  FID  | $F_8$ | $F_{\frac{1}{8}}$ |
> > > |-----------------|-------|-------|-------|-------------------|
> > > |   **Cifar10**   |       |       |       |                   |
> > > |     Config1     |  7.32 | 35.70 | 0.89  |        0.93       |
> > > |     Config2     |  8.14 | 16.28 | 0.96  |        0.97       |
> > > |     Config3     |  9.03 |  7.76 | 0.98  |        0.99       |
> > > |  **Animeface**  |       |       |       |                   |
> > > |     Config4     |  2.41 | 27.34 | 0.77  |        0.92       |
> > > |     Config5     |  2.18 |  8.81 | 0.94  |        0.98       |
> > > | Ours(remove BN) |  2.18 | 7.49  | 0.96  |        0.99       |
> > >
> > > * *Config1* is the default setting of the Phil Wang implementation, except batch size is 64 and image size is 32.
> > >
> > > * *Config2* is the setting of config1 except Unet dim is 128, dim_mults is [1,2,2,2], diffusion's loss type is 'l2' and learning rate is 2e-4.
> > >
> > > * *Config3* uses the Kim Seonghyeon implementation with the [setting of the original paper] (https://github.com/hmdolatabadi/denoising_diffusion/blob/master/config/diffusion_cifar10.json), except the batch size is 64 and n_iter is 500,000.
> > >
> > > * *Config4* is the sample implementation and setting of Config1, except the image size is 64.
> > >
> > > * *Config5* uses the Kim Seonghyeon implementation with [the default settings for CelebA](https://github.com/hmdolatabadi/denoising_diffusion/blob/master/config/diffusion_celeba.json) except the resolution is 64 and n_iter is 300,000.
> > >
> > > [1] Y.  Song  and  S.  Ermon.   Improved  techniques  for  training  score-based  generative  models. Advances in Neural Information Processing Systems (NeurIPS), 2020.

---

> > > > ### Comment · Area_Chair_92NE · 2021-09-02
> > > > **Change-of-variables formula**
> > > >
> > > > I would also like to ask something about Eq. (8) in the simplest case. The paper states that the dimension of $z$ (the noise variable) is $d$ and the dimension of $x=G(z)$ (in sample space) is $D$, and I believe $D>d$. Under this setting, consider the simplest case, a linear transformation $x=G(z)=Wz$, where $W\in\mathbb{R}^{D\times d}$, and consider the Gaussian distribution $p(z)=\mathcal{N}(0,I)$. Then, the density of $x$ does not exist, i.e., $\mathcal{N}(0, WW^\top)$ is infinite (since the covariance matrix isn't full-rank). However, the expression $\log p(x) = \log p(z) - \frac{1}{2}\log\det(W^\top W)$ is not infinite --- but it is not valid here. The fact that $\textrm{rank}(W)=d$ does not save you here. Can the authors discuss this point?

---

> > > > > ### Author Response · Authors · 2021-09-02
> > > > > **Re: Change-of-variables formula**
> > > > >
> > > > > Thank you for the question -- we are thankful for the engagement.
> > > > >
> > > > > Eq. 8 should be read as the expression of the density *where it has support*.
> > > > >
> > > > > Let us first consider the linear case. Assume $z \sim \mathcal{N}(0, I)$ with $z \in \mathbb{R}^d$,
> > > > > and $W \in \mathbb{R}^{D \times d}$ with $D > d$. Then we would write
> > > > > $x = W z \sim \mathcal{N}(0, W W^{\top})$ to indicate the distribution of $x$.
> > > > > As you say, this is a degenerate Gaussian as the covariance is not full rank.
> > > > > This is a valid density, but not on $\mathbb{R}^D$, but rather on the
> > > > > $d$-dimensional hyperplane spanned by $W$. The expression $x = W z \sim \mathcal{N}(0, W W^{\top})$
> > > > > should, thus, be read as the density expressed over this hyperplane. Outside the
> > > > > hyperplane, the density is $0$. See [this page](https://en.wikipedia.org/wiki/Multivariate_normal_distribution#Degenerate_case)
> > > > > for a detailed discussion.
> > > > >
> > > > > Note that Eq. 8 correctly captures this situation. For a point $x = W z$ which is
> > > > > on the hyperplane spanned by $W$, the density evaluates to
> > > > >
> > > > > $$
> > > > > \log p(W z) = \log p_0 (z) - \frac{1}{2} \log\det(W^\top W)
> > > > > $$
> > > > >
> > > > > The fact that the equation is only valid for points on the hyperplane spanned by $W$
> > > > > is only implicitly appearing in Eq. 8 where we write $G(z)$ instead of $x$, such
> > > > > that the equation is inapplicable to points that are not on the hyperplane.
> > > > > We admit that this could have been written more clearly, and we will adjust our writing.
> > > > >
> > > > > In the nonlinear case where $G: \mathbb{R}^d \rightarrow \mathbb{R}^D$ spans
> > > > > a nonlinear manifold (due to the full-rank assumption), then Eq. 8 gives an expression
> > > > > for the density, but only for points on the $d$-dimensional manifold spanned by $G$.
> > > > > Outside this manifold, the density is zero. This last point is not explicitly
> > > > > stated in the paper, which we will be happy to rectify.

---

> > > > > > ### Comment · Area_Chair_92NE · 2021-09-03
> > > > > > **Thank you - this is clear to me now**
> > > > > >
> > > > > > Thank you for your answer, this clarifies my question.

---

### Official Review · Reviewer_y6Eh · 2021-07-14

**Rating:** 8
**Confidence:** 4

**Summary:**

This paper proposes a method to train EBMs with lower and upper variational bounds. Previous approaches used only one side of the bound. This causes significant training instabilities as the training objective involves alternately maximizing and minimizing the same quantity.

The lower bound is estimated by lower-bounding the log determinant of the jacobian multiplied with itself with the smallest singular value of the jacobian.

The upper bound is inspired by tricks used to make WGAN and related EBM training approaches stable by regularizing the gradient.

The minimum singular value, needed for computing the lower bound, is computed using an iterative algorithm run for a small number of steps. Automatic differentiation is used cleverly to ensure full Jacobians don’t need to be instantiated.

The upper bound is loosened, and Hutchinson’s trace estimator is employed to efficiently compute.


**Limitations And Societal Impact:**

The authors have adequately addressed the limitations and potential negative societal impact of their work.

**Main Review:**

Describe the strengths of the work:

The authors develop a fast and stable approach to optimize EBMs like a GAN. They compare to strong baselines from both the EBM and GAN literature. They compare models on large image datasets. The authors perform an important ablation where they remove their upper bound and use the simpler gradient penalty method.

The authors develop a theoretical justification for a previous heuristic that was found empirically to significantly stabilize training.

The authors investigate the utility of their approach on a number of tasks including toy data, where density modelling can be compared exactly, mode coverage on MNIST, and sample quality and OOD on image data.

The authors give an explanation of some future work and open questions about their method.

Explain the limitations of this work:

The authors claim that their approach is better because it doesn’t require additional networks or rely on hard-to-tune parameters. It might be interesting to scale their approach up to larger images, for example 128x128 or even 256x256 images, but I understand this is difficult especially given the compute used as stated in the paper.

Correctness:

The authors carefully empirically verify a number of their claims. In Figure 3, they show the quality of their entropy estimator on a tractable problem.

Clarity:
The writing is clear.

Relation to prior work:

This work clearly describes how it builds on existing work by training models with a bidirectional bound, while previous approaches used only a bound in one direction, and a gradient regularization.

Reproducibility: Are there enough details to reproduce major results of this work

Details of the architecture, optimization, and other hyperparameters are provided in the appendix.

Additional feedback, comments, suggestions:

Please include details of data processing the appendix. For example, is the data normalized to be in [0, 1] or [-1, 1]? Was data dequantization used? What augmentations were used? What were the train-test splits used for optimizing models?

Why does the rank-order of timings in Table 5 change between the two experimental setups?


**Time Spent Reviewing:**

2

---

> ### Author Response · Authors · 2021-08-10
> **Reply**
>
> Thank you for the kind remarks. We agree that scaling the method to higher dimensional data is very interesting, but, as you say, this comes with an increasing computational demand. There should be no theoretical issues with scaling to higher dimensional data, though. We will comment on this in the paper.
>
> Regarding your questions:
>
> - We normalize the data to be in [-1, 1] and do not use dequantization.
> - During training we augment only using random horizontal flips.
> - For cifar10 we used the official train-test split from PyTorch, and for animeface we used a 85:15 train-test split.
>
> We will add this missing information to the supplementary appendix.
>
> Regarding the rank-order timings, then it turns out we had a mistake. One job had been scheduled to run on a different type of GPU than expected, which gave an incomparable timing. Thank you for catching this. In general, our method is 2 times slower than WGAN-0GP and MEG.

---

> > ### Comment · Reviewer_y6Eh · 2021-08-25
> > **Reply**
> >
> > The authors adequately addressed the small technical concerns I raised involving details of the data processing, the timing comparison, and scaling the approach to larger data.
> >
> > Additional concerns were raised by other reviewers, which I found the authors to adequately address.
> >
> > The discussion and comparison to NSCN and DDPM requested by reviewer xLGS were thoroughly addressed by the authors in my opinion. Additionally, I agree with the authors that score-based and diffusion models are entirely separate approaches from the methods discussed in the authors’ paper, and thus the comparison is not crucial. For example, EBMs as discussed in this work have applications outside of generating high-quality samples, as seems to be the focus of NSCN and DDPM and other works in that area. EBMs have applications to hybrid discriminative-generative modelling as explored in Grathwohl et al. (2021), OOD, and other applications. NSCN and DDPM focus mostly on sample quality, and appear to be much more computational expensive approaches (both in terms of training and sample generation) than EBMs.
> >
> > The concerns regarding local injectivity of the generator are adequately addressed in my opinion.
> >
> > I agree with the author’s response to rhEU regarding the justification for gradient penalty in the EBM literature.
> >
> > I strongly disagree with vzR3 comments regarding the related work. While perhaps some additional works should be included in the citations, I don’t think this is a severe gap in the paper and I don’t think it warrants such a low score.
> >
> > For these reasons I am raising my score to 8.

---

### Official Review · Reviewer_vzR3 · 2021-07-15

**Rating:** 5
**Confidence:** 5

**Summary:**

This paper proposes a bidirectional bound on the EBM log-likelihood, such that it can maximize a lower bound and minimize an upper bound when solving the minimax game of EBM. The paper demonstrates the effectiveness of the proposed method on density estimation and sample generation. The major contribution of the current paper is to use the bidirectional bounds to train the EBMs.

**Limitations And Societal Impact:**

Given the fact that the authors are not aware of the related literature and lack to compare their models to those closely related works. This might lead to a potential negative ethical impact.  I recommend a resubmission with a rejection.

To better understand the development and the advance of the EBMs, I suggest the authors review a new tutorial on EBMs in CVPR 2021 for a complete list of references, since the references I pointed out here are still not complete. This will be helpful to improve the current manuscript.


**Main Review:**

The paper addresses an important problem of training EBMs by using the bidirectional bounds to train the EBMs. The paper is easy to follow. And the results look promising.  The major contribution of the current paper is the usage of the bidirectional bounds.

However, the connection of the paper to the existing prior works and pioneering works is pretty bad. A lot of related works are missing and a necessary comparison with some baselines methods is also missing.  Related works are pointed out below.
(1)	About deep generative EBM (from the second paragraph in the related work). There has been a long history for generative EBM for image modeling and generation. For example, [1] is the first EBM with MCMC, i.e., Gibbs sampling, for texture images. [2] is the first EBM using Langevin dynamics for texture images. [3] and [4] are EBMs using Gibbs and HMC for object images. The EBMs proposed in [1][2][3][4] are not deep model but can be still considered a two-layer CNN (because convolution or filtering is used). Of course, it is OK for the current paper to not connect them because they are not DEEP generative EBM.  However, the pioneering work that uses a modern deep neural network to parameterize the EBM and uses Langevin dynamics-based MLE to train the EBM is [5], and [6][7] have even generalized to video and voxel domains. The earliest deep generative EBM references cited in the current paper are from 2019, e.g., Du 2019 and Grathwohl 2021. It is not OK to skip the first EBM with deep CNN.

[1] Filters, random fields and maximum entropy (FRAME): Towards a unified theory for texture modeling. IJCV, 1998.

[2] Grade: Gibbs reaction and diffusion equations. ICCV 1998.

[3] Inducing Wavelets into Random Fields via Generative Boosting. Journal of Applied and Computational Harmonic Analysis (ACHA) 2015

[4] Learning Sparse FRAME Models for Natural Image Patterns. International Journal of Computer Vision (IJCV) 2014

[5] A Theory of Generative ConvNet (ICML 2016)

[6] Synthesizing Dynamic Pattern by Spatial-Temporal Generative ConvNet (CVPR 2017)

[7] Learning Descriptor Networks for 3D Shape Synthesis and Analysis. (CVPR 2018)

(2)	About adversarial interpretation of EBM learning. The paper mentioned the minimax game of EBM or adversarial interpretation of EBM learning, but failed to cite the right paper or the first paper that pointing out this. [6] in 2017 is the first one to point out that. [7] and [8] also mentioned the adversarial game of EBM.

[8] Sparse and Deep Generalizations of the FRAME Model. Annals of Mathematical Sciences and Applications 2018

(3)	About learning a generator as an approximate sampler for EBMs (from the third paragraph in the related work). The current paper missed to discuss cooperative learning or CoopNet [9][10][11][12], which is about the joint training of EBM and top-down generator. In the cooperative training, the generator serves as an effective proposal distribution to efficiently evaluate the EBM. The current paper is related to the cooperative learning idea but didn’t cite or discuss it.

[9] Cooperative learning of energy-based model and latent variable model via MCMC teaching. AAAI 2018

[10] Cooperative learning of descriptor and generator networks. PAMI 2018

[11] Learning Energy-Based Model with Variational Auto-Encoder as Amortized Sampler. AAAI 2021

[12] Cooperative Training of Fast Thinking Initializer and Slow Thinking Solver for Conditional Learning. (arXiv 2019 or PAMI 2021)

(4)	About baselines. Since the proposed model is close to cooperative learning. The cooperative learning [9] that trains an EBM with a generator and the EBM-VAE [11] that trains an EBM with a VAE sampler should be compared in your experiments.


After Rebuttal:

In the rebuttal stage, authors made efforts to fill the gap between the current paper and the prior works. I updated my score from 3 to 5.  I urge the authors to substantially revise their paper by including the discussion and comparison in their reply. To be specific, (1) in your introduction, build a connection with pioneering works about joint training of EBM and generator. (2) complete the related work parts for both single EBM and joint training of EBM and generator. (3) include the quantitative comparison with the baselines, such as CoopNets, etc in experiments.



**Time Spent Reviewing:**

48h

---

> ### Author Response · Authors · 2021-08-10
> **Reply**
>
> Thank you for the feedback. When we review, we often find ourselves complaining about authors only citing very recent work, so it is very frustrating to us to have made the same mistake. We opted to cite surveys (LeCun et al., 2006 and Scellier, 2020) for early work on EBMs and refer to the review in the paper by Grathwohl et al. (2021) for the latest developments in deep EBMs. Still, you are correct that we should have included a more complete picture of the literature, and, in particular, we should direct reference the earliest work on deep EBMs as you have pointed out. This is a mistake on our part, but one that is easily fixable: we will extend Section 4 ('Related work') near the end of the paper to cover a wider selection of work. Note that this does not impact the novelty of our contribution as prior work does not study the idea of bidirectional bounds. As such, the changes needed to the paper are confined to Section 4 and are in some sense minimal.
>
> Regarding the proposed baselines, then Grathwohl et al. (2021) compares VERA to the 2018 paper you mention (Xie et al., 2018) and find that VERA performs significantly better. Since we, in turn, perform better than VERA, we have decided not to include this additional baseline. The 2021 paper (Xie et al., 2021) you request as a baseline was published one week before the Neurn PS deadline, so we honestly do not consider this request to be entirely fair.

---

> > ### Comment · Reviewer_vzR3 · 2021-08-19
> > **Adding back prior works is easy, but evaluating a paper without mentioning prior close works is risky.  Major revision is needed before re-rating.**
> >
> > Thanks for your reply.
> >
> > I agree that completing the related works by adding necessary references is easy to address, however, the lack of discussion of related work might affect the judgment of the reviewers on the novelty of the paper, and might lead to potential negative societal impact or ethical issue that have been brought into the community’s attention. We should not underestimate this issue. If the related works, especially the close works, are not introduced or discussed, reviewers are hard to justify the significance of the paper.
> >
> > As to the prior works of EBMs: Survey (LeCun et al., 2006) is too old and it mainly focuses on discriminative EBMs but generative EBMs.   Grathwohl et al. (2021) is a new publication but it doesn’t cover a necessary list of pioneering works about ConvNet-EBM, for example, it even doesn’t cite [xie et al. 2016], which is the first deep EBM with Langevin sampling, not to speak of other important works. Therefore, it can not serve as a representative survey for current EBMs.
> >
> > As to the works that are very close to the current paper:
> > The current paper mentioned that in line 144 “Our proposed method belongs to the fourth class of algorithms that sidestep the costly MCMC sampling allowing fast sampling”. The fourth class of generative models is called “cooperative learning” or “generative cooperative network (CoopNets)” in [Xie 2018, AAAI and PAMI, or ArXiv 2016], which is the earliest and formally published work. The cooperative network includes an EBM and a generator, where the generator is a fast sampler to help EBM for MCMC. Other variants that follow the same spirit actually belong to cooperative networks. Actually, this framework has been used in other applications, has other variants, and could not be ignored or neglected in this paper. For example, [9]-[14]. Without discussing those existing works that are close to the current paper or belong to the same family, reviewers are hard to understand the significance of the current paper. Because all reviewers, authors, and ACs should take responsibility for that. We need a full understanding of the difference between the existing works and the current work before we can make a correct evaluation.  Adding back prior works is easy, but evaluating a paper without mentioning prior close works is risky.
> >
> > As to baselines: As I mentioned before, the “fourth-class” generative models (cooperative networks) include several existing models, but the current paper only compares with VERA [Grathwohl et al. (2021)], which is also a variant of CoopNets. I think the authors might not be aware of other baselines or similar works, which means the current paper might have a chance to bear some similarities or novelties with existing works that they might not know. I suggest that authors should take action to go over the models in the “fourth class” and improve the clarity of the paper, which seems to be more important than others. Even though the VERA paper shows that its performance is better than that of (CoopNets)” [Xie 2016, 2018], this is not the reason that we can always compare with the STOA. I believe, most of the published papers do not just compare with a single method, which is the STOA. And more importantly, I found that the comparison with CoopNets in the VERA paper was added during their rebuttal (I checked the ICLR forum of that paper), the authors claimed that they didn’t carefully tune the CoopNets to get more reasonable results because the rebuttal time is limited, thus I don’t believe VERA is certainly better than CoopNets in any cases. Again, even if GANs always outperform VAEs in most of the papers, we should not say, we will not compare VAE anymore. Again and Again, VERA is not the only model in the ‘’’fourth class’.  I expect a more comprehensive comparison with other existing models that follow the same spirit as your work. Also, the 2021 paper [Xie et al., 2021] actually is an arXiv Dec 2020 paper, which is before the NeurIPS 2021 submission deadline. I believe there are more related baselines that are not listed below that need to be involved in your experiments and discussion.
> >
> > Given the fact that the current paper needs a serious major revision. And I can not update my score before seeing the new version of the paper. I have to keep my current rating and urge the authors to improve the paper by taking all reviews' feedbacks.
> >
> > =============
> >
> > References
> >
> > [9] Cooperative learning of energy-based model and latent variable model via MCMC teaching. AAAI 2018
> >
> > [10] Cooperative learning of descriptor and generator networks. PAMI 2018
> >
> > [11] Learning Energy-Based Model with Variational Auto-Encoder as Amortized Sampler. AAAI 2021
> >
> > [12] Cooperative Training of Fast Thinking Initializer and Slow Thinking Solver for Conditional Learning. (arXiv 2019 or PAMI 2021)
> >
> > [13] Energy-Based Generative Cooperative Saliency Prediction. arXiv 2021
> >
> > [14] Learning Cycle-Consistent Cooperative Networks via Alternating MCMC Teaching for Unsupervised Cross-Domain Translation. AAAI 2021

---

> > > ### Author Response · Authors · 2021-08-19
> > > **Follow-up**
> > >
> > > We agree that the discussion of related work is important for the review process and that this is a delicate issue as relevant papers are almost always omitted.
> > >
> > > However, we think the reviewer takes this viewpoint too far.
> > >
> > > ## Citation requests
> > >
> > > Indeed, we did not cite any of the 14 papers that all are by a narrow group of authors, and we acknowledge that this was a mistake. Our key contribution is the idea of bi-directional bounds, an idea that has not been explored in previous work. And indeed, much of the work you request us to cite is requested for historical accuracy rather than novelty.
> > >
> > > Also, the reviewer writes,
> > > > the current paper might have a chance to bear some similarities or novelties with existing works that they might not know
> > >
> > >  but does not provide any examples of works that do bear such similarities. If you want to reject our paper based on overlap with other work, it seems fair to expect you to point to this work.
> > >
> > > ## Comments on our work
> > >
> > > The reviewer's only assessment of our actual work reads,
> > >
> > > > The paper addresses an important problem of training EBMs by using the bidirectional bounds to train the EBMs. The paper is easy to follow. And the results look promising. The major contribution of the current paper is the usage of the bidirectional bounds.
> > >
> > >  The remaining comments pertain to missing citations of papers, which we are happy to add.  We find that the exclusive focus on the citations is unfair to the contributions of our paper.  We find it most unfortunate that the reviewer does not engage in a discussion of the actual merits of our work. Raising this discussion to the level of *"negative societal impact or ethical issue[s]"* is excessive.
> > >
> > > ## Baselines
> > >
> > > Regarding baselines, then there is always a curation at play. We find that with the additional experiments provided in our replies to other reviewers, we have quite good coverage. A fair comparison to Xie et al. is non-trivial as it relies on Langevin sampling, which is notoriously difficult to tune (also evident from the VERA rebuttal to which you referred). We can compare in terms of FID and IS scores on cifar10 as Xie et al. [2018a, 2018b, 2021] provide such a study; here we outperform Xie et al. (see table below).
> > >
> > > We further note that the 2021 paper by Xie et al. you requested a comparison with has no public implementation. Asking us to compare empirically with a paper published one week before the NeurIPS deadline and which has no public implementation based on the argument that a *preprint* was on arXiv is beyond unreasonable.
> > >
> > > |            Model           |              IS             |              FID             |
> > > |----------------------------|-----------------------------|------------------------------|
> > > |           Cifar10          |                             |                              |
> > > | CoopNets(Xie et al. 2018a) |             6.55            |             36.4             |
> > > | CoopNets(Xie et al. 2018b) |              -              |             33.61            |
> > > |  EBM-VAE(Xie et al. 2021)  |             6.65            |             36.2             |
> > > |            Ours            | $\textbf{7.45 $\pm$ 0.014}$ | $\textbf{28.63 $\pm$ 0.290}$ |

---

> > > > ### Comment · Reviewer_vzR3 · 2021-08-29
> > > > **Follow-up**
> > > >
> > > > I just point out that “The lack of discussing important related works **might** lead to potential negative societal impact, which means we should not overlook this issue, **sometimes** it is not just an issue of citation or adding references, but academic ethics. Also, I am NOT saying that the current paper under review is experiencing this situation.
> > > >
> > > > I appreciate that the authors are willing to involve the discussion of those related works and add a preliminary comparison with some baselines. This will certainly improve the paper in terms of completeness.
> > > > More importantly, what I expect is not just a simple citation but an insightful discussion of the difference, similarity, and connections of the current model and other members in the “fourth class of algorithm (or cooperative learning)”, such as [9-11], Kim and Bengio (2016), Zhai et al. (2016), and VERA. For example, is your learning equivalent to each of them? Why is your solution better than them? Are your EBM and generator play the same roles as those in other frameworks? That is, citing them is not enough. I expect an insightful understanding of the relationship among them to better understand the contribution. This part is unclear to me.
> > > >
> > > > Last but not least, about "EBM has an adversarial interpretation or a mini-max form".  [6] is the first one to show that. As to no MCMC to train EBM and generator, [15] is also a baseline, which is earlier than VERA. Check Equation 17 in paper [15], and compare it with your equations (5) and (6). Your paper is much closer to [15] than W-GAN as to the loss, because at least [15] uses an EBM and a generator, but W-GAN doesn’t have an EBM.
> > > >
> > > > [6] Synthesizing Dynamic Pattern by Spatial-Temporal Generative ConvNet (CVPR 2017)
> > > >
> > > > [15] Divergence triangle for joint training of generator model, energy-based model, and inference model. (CVPR 2019).

---

> > > > > ### Author Response · Authors · 2021-09-01
> > > > > **Follow-up**
> > > > >
> > > > > Thank you for the clarifying follow-up. Below we give a detailed discussion of similarities and differences between our work and the requested references. We will include a paragraph in the paper summarizing this discussion, but we include a long discussion here for distinctness. We note that at a high level, our work is different from the other work by being the first to study the use of bi-directional bounds.
> > > > >
> > > > > ### Cooperative learning of energy-based model and latent variable model via MCMC teaching. Xie et al., AAAI 2018 [9].
> > > > >
> > > > > **Similarities:**
> > > > > Xie et al. use a *descriptor* (akin to a *discriminator*) network as an energy function and a generator network to synthesize examples from the latent variables. They use the energy distribution as a teacher to guide the training of the generator.
> > > > >
> > > > > **Differences:**
> > > > > The method by Xie et al. (2018) is not based on an adversarial strategy. It uses Langevin sampling to draw from the energy distribution, which is expensive and difficult to tune. The generator is modeled with a stochastic function rather than our deterministic choice. Also, the objective is to maximize the data log-likelihood of the generator distribution, which is different from our bounds. Arguably, maximum likelihood is to be preferred over using bounds, but the price is the difficult and expensive reliance on Langevin sampling. Our bi-directional bounds are, thus, a more practical approach.
> > > > >
> > > > > ### Cooperative learning of descriptor and generator networks. Xie et al., PAMI 2018 [10]
> > > > >
> > > > > This is the journal version of [9] with more experiments, and the model and algorithms are by large the same.
> > > > >
> > > > > ### Learning Energy-Based Model with Variational Auto-Encoder as Amortized Sampler. Xie et al., AAAI 2021 [11].
> > > > >
> > > > > **Similarities:**
> > > > > Xie et al. (2021) use an energy function to estimate the data density and a generator to draw samples. They also minimizes the KL divergence between the energy distribution and real data distribution and then minimize the KL divergence between the generator distribution and the energy distribution.
> > > > >
> > > > > **Differences:**
> > > > > This method is not based on the adversarial strategy. Like [9, 10] above, it uses Langevin sampling to draw samples from the energy distribution with the difficulties this entails. The generator uses variational MCMC teaching to make the KL divergence between the generator distribution and energy distribution tractable. As a consequence, it has an extra encoder network compared with our approach.
> > > > >
> > > > > ### Deep Directed Generative Models with Energy-Based Probability Estimation. Kim & Bengio, ICLR workshop 2016.
> > > > >
> > > > > **Similarities:**
> > > > > This method by Kim & Bengio (2016) also has an energy function and a generator function whose adversarial training strategy is similar to ours. They update the generator using the same lower bound as us by approximating the entropy of the generator distribution.
> > > > >
> > > > > **Differences:**
> > > > > This method does not have a gradient penalty like our upper bound when optimizing the energy function. The used loss function matches our Eq. 6. Consequently, the energy function needs to be explicitly designed to prevent it from growing to infinity, limiting its potential. Furthermore, the method to approximate entropy is quite different: Kim & Bengio use the entropy of the normal distribution after batch normalization over each activation which is not accurate.
> > > > >
> > > > > ### Generative Adversarial Networks as Variational Training of Energy Based Models. Zhai et al., arXiv, 2016.
> > > > >
> > > > > **Similarities:**
> > > > > The method proposed by Zhai et al. (2016) also plays a min-max game to optimize the energy function and generator. It also has a regularizer with the proposed bounded multi-modal energy to avoid the scale of energy exploding. It also uses the same lower bound as us by approximating the entropy term of the generator distribution.
> > > > >
> > > > > **Differences:**
> > > > > The method relies on a specific design of the energy function, which limits its potential. The approximation of the entropy has no theoretical guarantee to recover the true entropy to any extent. Its regularizer for the energy loss function is not an upper bound; it is just observed to be helpful by experience. For their VCD method, it does not have a strict theoretical guarantee for their parameterized variational distribution to approximate the true transition distribution. Their entropy term for VCD is also obtained from experience.
> > > > >
> > > > > ### No MCMC for me: Amortized sampling for fast and stable training of energy-based models. Grathwohl et al., ICLR 2021.
> > > > >
> > > > > **Similarities:**
> > > > > Like our work, VERA by Grathwohl et al. (2021) avoids the use of MCMC. VERA also plays a min-max game and uses variational inference for the approximation of the entropy term.
> > > > >
> > > > > **Differences:**
> > > > > This method uses a zero-centered gradient penalty as a regularizer for the energy function as this is found to be practically needed. This is a heuristic, unlike our upper bound which is justified by the model. VERA uses a variational approximation with importance sampling to estimate the gradient of the entropy term, which is different from us.
> > > > >
> > > > > ### Divergence triangle for joint training of generator model, energy-based model, and inference model. Han et al., CVPR 2019 [15].
> > > > >
> > > > > **Similarities:**
> > > > > Han et al. (2019) also use an adversarial learning strategy in its divergence triangle loss. They also use an energy function to estimate the data density and a generator to draw samples.
> > > > >
> > > > > **Differences:**
> > > > > The training mechanism is radically different from ours. It has an extra encoder for approximate variational learning of the generator distribution. Eq. 17, which you mention, has a form like our Eq. 6 rather than the upper bound that we optimize.
> > > > >
> > > > > ### Synthesizing Dynamic Pattern by Spatial-Temporal Generative ConvNet. Xie et al., CVPR 2017 [6].
> > > > >
> > > > > Kim & Bengio (2016) and Xie et al. (2017) simultaneously observed the adversarial interpretation of EBMs, and we are, of course, happy to cite both for this contribution.

---

> > > > > > ### Comment · Reviewer_vzR3 · 2021-09-02
> > > > > > **please revise your paper by including the discussion and comparison in your reply.**
> > > > > >
> > > > > > Thanks for the reply. I urge the authors to revise their paper by substantially including the discussion and comparison in your reply.
> > > > > >
> > > > > > To be specific, (1) in your introduction, please try to build a connection with prior works and grant credits to pioneering works about joint training of EBM and generator. (2) please complete your related work parts for both the paragraph of single EBM and the paragraph of joint training of EBM and generator.  (3) please include the comparison with the baselines, such as CoopNets, etc in your experiments to make the work more solid.
> > > > > >
> > > > > > I increase my score from 3 to 5, to vote for it as a borderline paper by assuming all the above revisions are well taken. I will leave the final decision to AC. I will not be the blocker of this paper if it is recommenced to be accepted.

---

### Decision · Program_Chairs · 2021-09-27

**Decision:**

Accept (Poster)

**Comment:**

The main contribution of the paper is the development of lower and upper bounds for training EBMs in a stable manner, by drawing inspiration from the WGAN objective. The methodology presented in the paper can be useful for the research community. The reviewers raised some concerns about the paper that were adequately addressed during the rebuttal phase. I therefore recommend the paper for acceptance, but strongly encourage the authors to address the following points for the camera-ready version:

+ Rephrase the points about the instability & grad clipping of WGAN, as discussed with reviewer rhEU.

+ Add the experiment details to the appendix, as suggested by reviewer y6Eh.

+ Add some discussion and references in the related work section. In particular, it'd be useful to include as much of the similarities and differences with previous methods (from the response to reviewer vzR3) as possible.

+ Add some clarification about the invertibility of the Jacobian determinant and the change-of-variables formula, as discussed with reviewer xLGS and myself. In particular, the assumptions under which Eq. (8) holds should be clearly stated in the paper - these are pointed out by the authors in their rebuttal, but some additional questions are: Does this limit the architecture of the generative NN? For example, does each hidden layer need to have a number of hidden units between $d$ and $D$, with non-decreasing sizes along the layers? Is any non-linearity valid? (e.g., ReLu's map many input values to 0, so it shouldn't be used). If any assumption is made, it should be clearly stated in the final version. In addition, the text should clarify that $p(G(z))$ in Eq. (8) is *not* the Lebesgue measure on $\mathbb{R}^D$ but it is restricted to some low-dim manifold.

+ Incorporate the experiments that were ran during the rebuttal period.

Besides the items above, I also have two technical questions that should be clarified in the paper:

+ Below Eq. (20), the paper says that $\zeta=1$ allows for the simplified bound from Eq. (15). Shouldn't it be $\zeta=M$ instead, which corresponds to $m=1$ in Eqs. (13)-(15)?

+ Theorem 2 guarantees the existence of constants ($m$, $M$, and $p\geq 1$) such that the bound in Eq. (13) holds, but unless I'm missing something, it doesn't say what values of the constants guarantee the bound in Eq. (13). But, according to the rebuttal, Eq. (15) is a valid bound for any $p\geq 1$ (as long as the "expression$\geq 1$" assumption is met). So, even though it was obtained from Eq. (13), Eq. (15) makes an explicit statement about what values of the constants constitute a valid bound. How is such discrepancy possible? Is the "expression$\geq 1$" assumption the explanation for that? Does Theorem 2 actually say something about the values of the constants that guarantee the correctness of the bound?